# Sub-nanomolar sensitive GZnP3 reveals TRPML1-mediated neuronal $Zn^{2+}$ signals

Taylor F. Minckley [1], Chen Zhang[1], Dylan H. Fudge[1], Anna M. Dischler[1], Kate D. LeJeune[1], Haoxing Xu [2] & Yan Qin [1]*

Although numerous fluorescent $Zn^{2+}$ sensors have been reported, it is unclear whether and how $Zn^{2+}$ can be released from the intracellular compartments into the cytosol due to a lack of probes that can detect physiological dynamics of cytosolic $Zn^{2+}$. Here, we create a genetically encoded sensor, GZnP3, which demonstrates unprecedented sensitivity for $Zn^{2+}$ at sub-nanomolar concentrations. Using GZnP3 as well as GZnP3-derived vesicular targeted probes, we provide the first direct evidence that $Zn^{2+}$ can be released from endolysosomal vesicles to the cytosol in primary hippocampal neurons through the TRPML1 channel. Such TRPML1-mediated $Zn^{2+}$ signals are distinct from $Ca^{2+}$ in that they are selectively present in neurons, sustain longer, and are significantly higher in neurites as compared to the soma. Together, our work not only creates highly sensitive probes for investigating sub-nanomolar $Zn^{2+}$ dynamics, but also reveals new pools of $Zn^{2+}$ signals that can play critical roles in neuronal function.

[1] Department of Biological Sciences, University of Denver, Denver, CO 80210, USA. [2] Department of Molecular, Cellular, and Developmental Biology, University of Michigan, 4114 Biological Sciences Building, 1105 North University, Ann Arbor, MI 48109, USA. *email: yan.qin@du.edu

Zinc is one of the most abundant trace elements and bioinformatic studies have suggested that around 2800 human proteins have $Zn^{2+}$ coordination sites[1]. Though the majority of intracellular $Zn^{2+}$ is in the protein-bound state, it also exists in an unbound, labile state. Cells maintain a baseline labile $Zn^{2+}$ concentration of roughly 100 pM in the cytosol, and a high concentration pool of labile $Zn^{2+}$ might be stored within vesicular compartments, including lysosomes[6,7] and synaptic vesicles[8]. It is well accepted that such vesicular $Zn^{2+}$ in secretory cells (neurons[9], prostate[10], crypts of Lieberkühn[11], pancreatic beta cells[12]) can be released extracellularly by exocytosis to execute modulatory effects between cells. However, it has been presently uncharacterized if and how these pools can be accessed within the cell.

Electrophysiological evidence found that Transient Receptor Potential Mucolipin 1 (TRPML1) is a non-selective cation channel permeable to $Ca^{2+}$[13], $Fe^{2+}$[14], $Mn^{2+}$[15], and $Zn^{2+}$[14]. Mutations in TRPML1 are the genetic cause of the disease Mucolipidosis type IV (MLIV). The most prominent symptoms of MLIV are severe neurological underdevelopment and neurodegeneration. Extensive studies have focused on TRPML1-mediated $Ca^{2+}$ signals that might have potential biological roles in lysosomal motility[16] and autophagy[17]; however, the roles of TRPML1 in neuronal function are still unclear. Given that TRPML1 is localized on lysosomes[14,18–21] and TRPML1 knockdown causes lysosomal $Zn^{2+}$ accumulation[22,23], we hypothesized that TRPML1 might mediate $Zn^{2+}$ release from lysosomes to the cytosol.

Direct tracking of TRPML1-mediated $Zn^{2+}$ signals requires probes that are able to monitor real time sensitive dynamic signals in living cells. The sensitivity of a biosensor is defined as the ratio of the changes in sensor output $[(F-F_0)/F_0]$ to changes in $Zn^{2+}$ concentration $(\Delta[Zn^{2+}])$, thus the ideal sensor sensitivity is dependent on the range of $Zn^{2+}$ concentrations of interest. We predicted that opening of TRPML1 channels can release vesicular $Zn^{2+}$ into the cytosol and might induce an elevation of local $Zn^{2+}$ concentrations from the picomolar to nanomolar range (100 pM–1 nM). Thus we need sensors that can detect $Zn^{2+}$ with sub-nanomolar sensitivity.

To assess sensor sensitivity, we take into account the sensor's apparent dissociation constant $(K_d)$ for $Zn^{2+}$, in situ dynamic range (maximum to minimum signal ratio), and in situ maximal sensor response above baseline. When $Zn^{2+}$ concentration changes within 10-fold below and above the $K_d$, the sensor demonstrates the most significant changes[24]. As such, we defined a sensor's sensitive range as the concentration from 10% to 10-fold of $K_d$. Most small molecule sensors possess high $K_d$, limiting their sensitivity for $Zn^{2+}$ changes between 100 pM and 1 nM[25–28]. For example, the widely utilized small molecule sensor FluoZin-3 (Thermofisher) presents a large dynamic range but a $K_d$ of 15 nM, confining its sensitive detection to $Zn^{2+}$ concentrations within the nanomolar range (~1.5–150 nM). For all previously published genetically encoded $Zn^{2+}$ sensors, eight sensors display compatible binding affinity in the sub-nanomolar range, but their sensitivities suffer from low dynamic range (Table 1). For all the FRET sensors, their maximal response to $Zn^{2+}$ saturation yields small signals (0–0.65) above or below the baseline in cells (Table 1), preventing them from sensitively detecting the changes in $Zn^{2+}$ concentrations at sub-nanomolar to nanomolar levels.

In this manuscript, we describe the development of a novel, sensitive fluorescent protein-based $Zn^{2+}$ probe, GZnP3, capable of detecting sub-nanomolar $Zn^{2+}$ dynamics in live cells. Using this genetically encoded sensor, which can be targeted to various subcellular compartments, we directly demonstrate for the first time that TRPML1 is permeable to physiological concentrations of $Zn^{2+}$, and that its activation in hippocampal neurons mediates the release of $Zn^{2+}$ from late endosomes and lysosomes to the cytosol. Additionally, TRPML1-mediated $Zn^{2+}$ release is unique to neurons and INS-1 cells, as compared to non-secretory

**Table 1 Comparison of GZnP3 with previously published genetically encoded $Zn^{2+}$ sensors**

| Sensor type | $Zn^{2+}$ sensor | Apparent dissociation constant $(K_d)$ | In situ dynamic range $(S_{max}/S_{min})$ | Maximal response above baseline $[(S_{Zn}-S_0)/S_0]$ | Ref |
|---|---|---|---|---|---|
| FRET | ZapCY1 | 2.5 pM[a] | 4.2 | 0 | 5 |
| **FRET** | **ZapCY2** | **811 pM[a]** | **1.4** | **0.33** | 5 |
| FRET | ZapCmR1.1 | n.d. | 1.4 | 0.38 | 61 |
| FRET | ZapCmR2 | n.d. | 1.2 | 0.33 | 61 |
| **FRET** | **ZapCV2** | **2.3 nM[a]** | **2.2** | **0.49** | 62 |
| FRET | ZapCV5 | 0.3 μM[a] | 1.5 | 0.36 | 62 |
| FRET | eCALWY-1 | 2 pM[b] | 1.6 | −0.07[c] | 3 |
| FRET | eCALWY-2 | 9 pM[b] | 1.6 | 0 | 3 |
| FRET | eCALWY-3 | 45 pM[b] | 1.6 | **−0.14[c]** | 3 |
| **FRET** | **eCALWY-4** | **630 pM[b]** | **1.7** | **−0.21[c]** | 3 |
| **FRET** | **eCALWY-5** | **1.8 nM[b]** | **1.3** | **−0.27[c]** | 3 |
| **FRET** | **eCALWY-6** | **2.9 nM[b]** | **1.3** | **−0.2[c]** | 3 |
| FRET | redCALWY-1 | 12.3 pM[b] | 1.2 | 0 | 63 |
| **FRET** | **redCALWY-4** | **234 pM[b]** | **1.2** | **−0.15[c]** | 63 |
| FRET | eZinCh-1 | 8.2 μM[b] | n.d. | n.d. | 3,64 |
| **FRET** | **eZinCh-2** | **1.0 nM[b]** | **2.6** | **0.65** | 27,65 |
| Single FP | ZnGreen1 | 633 nM[a] | 2.5 | 0 | 66 |
| Single FP | ZnGreen2 | 20 μM[a] | n.a | n.d | 66 |
| Single FP | ZnRed | 166 nM and 20 μM[a] | n.a | n.d | 28,66 |
| Single FP | GZnP1 | 34 pM[a] | 2.2 | 0.46 | 29 |
| **Single FP** | **GZnP2** | **352 pM[a]** | **4.5** | **2.5** | 2 |
| **Single FP** | **GZnP3** | **1.3 nM[a]** | **11** | **4** | |

Bold sensors show potential detection sensitivity between 100 pM and 1 nM. FRET sensors were measured using ratio of acceptor to donor emission $(\Delta R/R_0)$ and single FP sensors were measured using single emission $(\Delta F/F_0)$
$S_{max}$: sensor maximum signal. $S_{min}$: sensor minimum signal. $S_{Zn}$: $Zn^{2+}$-saturated signals. $S_0$: baseline cytosolic sensor signal
[a]The $K_d$ was measured at pH 7.4
[b]The $K_d$ was measured at pH 7.1
[c]Negative values indicate sensors with turn-off kinetics upon $Zn^{2+}$ binding

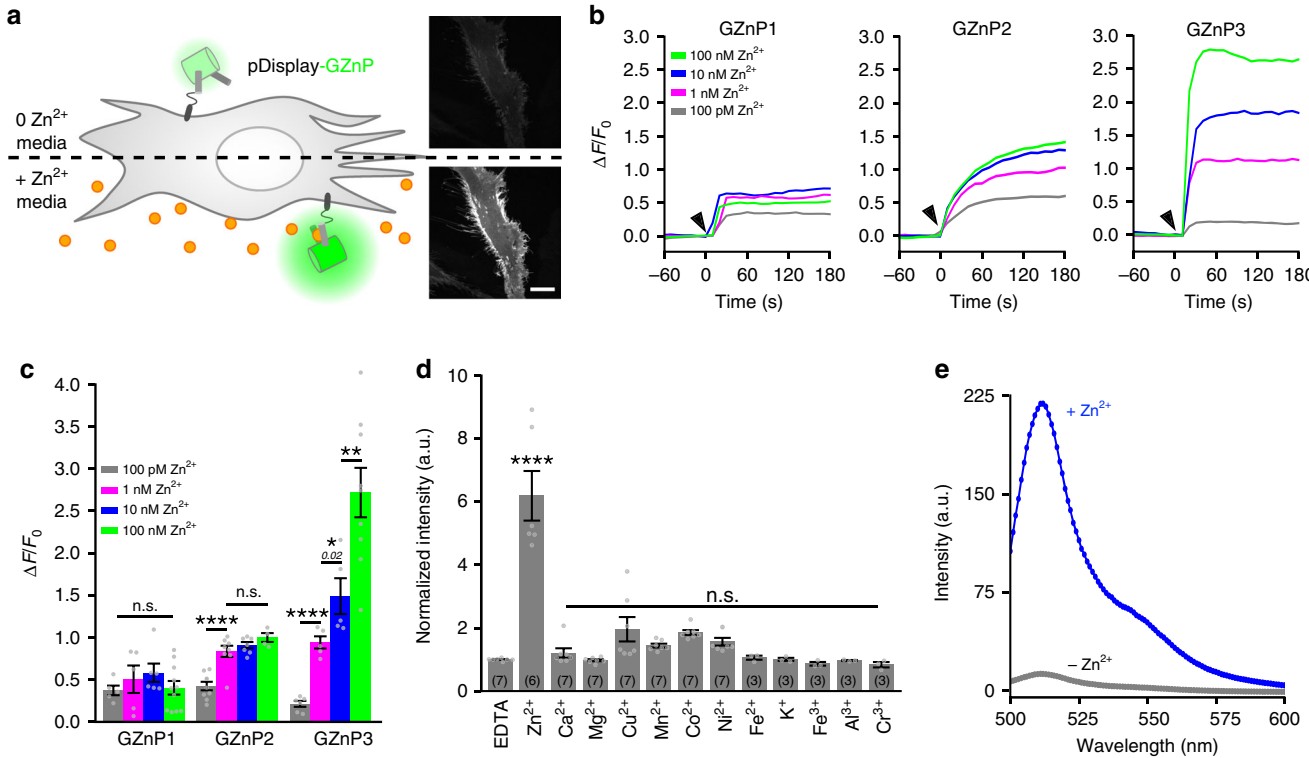

**Fig. 1** GZnP3 sensor development and characterization. **a** Schematic of the pDisplay assay showing fluorescence change of GZnP sensors in response to $Zn^{2+}$ (left). Representative micrographs illustrated the localization of GZnP3 sensor on extracellular face of plasma membrane in HeLa cells, where GZnP3 fluorescence increased with $Zn^{2+}$ (right). Scale bar = 20 μm. **b** Representative traces for each extracellular $Zn^{2+}$ concentration (gray, 100 pM; green, 1 nM; blue, 10 nM; red, 100 nM) for GZnP1, GZnP2, and GZnP3. Cells were pretreated with 100 μM TPEN, washed, then imaged in the presence of 500 μM TCEP. Buffered $Zn^{2+}$ solutions were added at 0 sec (black arrow). **c** Mean $\Delta F/F_0$ (± s.e.m.) at 60 s for GZnP1, GZnP2, and GZnP3. GZnP1: 100 pM ($n = 5$ cells), 1 nM ($n = 5$), 10 nM ($n = 6$), 100 nM ($n = 11$); GZnP2: 100 pM ($n = 9$), 1 nM ($n = 8$), 10 nM ($n = 7$), 100 nM ($n = 4$); GZnP3: 100 pM ($n = 6$), 1 nM ($n = 5$), 10 nM ($n = 5$), 100 nM ($n = 9$). Student's one-tailed $t$-tests. **d** In vitro metal specificity of GZnP3 for $Zn^{2+}$ as compared to EDTA control (12 μM), and a panel of biologically relevant divalent cations (50 μM each in the presence of 12 μM EDTA). Mean intensity ± s.e.m. (n wells displayed on bars), normalized to EDTA control. One-way ANOVA, with Dunnett's multiple comparison to EDTA control. **e** Representative in vitro emission spectrum of GZnP3 in the apo (gray) and the $Zn^{2+}$ saturated (blue) state when excited at 488 nm. GZnP3 displayed an emission peak at 512 nm and the peak fluorescence intensity increased 17-fold over the apo state in response to $Zn^{2+}$. $*p < 0.05$, $**p < 0.01$, $****p < 0.0001$, n.s. not significant. $p$-values displayed in italics above corresponding comparisons where applicable. Source data are provided as a Source Data file

mammalian cell lines. Most importantly, neurites exhibit greater (2.6-fold) $Zn^{2+}$ signals, but reduced $Ca^{2+}$ signals, than the soma, when TRPML1 is activated in neurons.

## Results

**Engineering GZnP3 sensor with sub-nanomolar sensitivity.** In order to generate a sensor with higher sub-nanomolar sensitivity than previous genetically encoded sensors, we chose the single-fluorescent protein-based platform as previously utilized to create GZnP1 and GZnP2 in our lab (Table 1)[2,29]. The GZnP family of $Zn^{2+}$ sensors function using a circularly permuted Green Fluorescent Protein (cpGFP) fused to the two $Zn^{2+}$-binding finger domains of yeast *Saccharomyces cerevisiae* transcription factor Zap1[2,29]. $Zn^{2+}$ binding can induce the conformational change of the two $Zn^{2+}$ fingers, resulting in an increase in the fluorescent intensity of cpGFP. Using a high throughput cell lysate screening assay as previously described[2] we identified several new sensor variants with an improved dynamic range.

To compare the sensor sensitivity among various GZnP variants and identify the sensor with sub-nanomolar sensitivity in live cells, we designed an in situ pDisplay assay to monitor and measure sensor responses to a range of defined picomolar to nanomolar $Zn^{2+}$ concentrations in intact cells (Fig. 1a). GZnP sensors cloned in frame with the pDisplay vector are secreted and

anchored on the extracellular side of the plasma membrane, allowing direct measurement of each sensor's response to known $Zn^{2+}$ concentrations. Compared to in vitro measurements with purified sensor protein, this assay provides a faster and reliable method to explore sensor sensitivity in live cells. Using HeLa cells expressing these GZnP sensor variants, we can monitor sensor response within seconds to a defined $Zn^{2+}$ concentration (100 pM, 1 nM, 10 nM, and 100 nM) by fluorescence microscopy (Fig. 1a–c). By this pDisplay assay, we identified a new sensor GZnP3 with enhanced sensitivity to sub-nanomolar $Zn^{2+}$. Compared to GZnP1 and GZnP2, the GZnP3 sensor clearly differentiated $Zn^{2+}$ concentrations from 100 pM to 100 nM (Fig. 1c) and demonstrated significantly higher fluorescence change than GZnP1 and GZnP2 in response to 1 nM, 10 nM, and 100 nM $Zn^{2+}$, separately. Such experimental results are consistent with our predictions based on $K_d$ in Table 1. GZnP1 showed no sensitivity between 100 pM to 1 nM because its sensitive detection range is restricted from ~3.4–340 pM. For GZnP2, the sensitive range is ~35 pM–3.5 nM, which agrees with the pDisplay results that GZnP2 yields the highest sensitivity between 100 pM and 1 nM $Zn^{2+}$ (Fig. 1b & c). The sensitive range of GZnP3 is similar to GZnP2, but its high dynamic range confers superior sensitivity than GZnP2. GZnP2 sensor response ($\Delta F/F_0$) increased from 0.42 to 0.83 in response to $Zn^{2+}$ changing from 100 pM to 1 nM, while GZnP3 sensor increased response from 0.21 to 0.94 (Fig. 1c).

Therefore, GZnP3 had a 2.25-fold greater response than GZnP2 to the same change in $Zn^{2+}$ concentration. In addition, GZnP3 showed a faster turn-on kinetics than GZnP2 in response to $Zn^{2+}$ (Supplementary Fig. 2b, c).

The biophysical features of GZnP3 sensors were revealed by in vitro characterization. GZnP3 showed turn-on fluorescence response to $Zn^{2+}$ in vitro with nanomolar affinity ($K_d = 1.3$ nM) (Fig. 1e, Supplementary Fig. 2a and Supplementary Table 1). The GZnP3 sensor was specific for $Zn^{2+}$ over other biologically relevant metals ($Ca^{2+}$, $Mg^{2+}$, $Cu^{2+}$, $Mn^{2+}$, $Co^{2+}$, $Ni^{2+}$, $Fe^{2+}$, $K^+$, $Fe^{3+}$, $Al^{3+}$, or $Cr^{3+}$) (Fig. 1d). Metal specificity was additionally investigated in HeLa cells to examine the sensor response to $Fe^{2+}$ and $Ca^{2+}$ that can also be transported by TRPML1. Upon addition of ionomycin, a $Ca^{2+}$ ionophore, in the presence of 1.26 mM extracellular $Ca^{2+}$, there was no observed changes in GZnP3 fluorescence, confirming that $Ca^{2+}$ does not affect baseline GZnP3 fluorescence (Supplementary Fig. 1a). Additionally, the $Fe^{2+}$ chelator, 2,2'-bipyridyl, showed no effects on baseline GZnP3 fluorescence (Supplementary Fig. 1b) while addition of 100 μM N,N,N′,N′-tetrakis(2-pyridinylmethyl)-1,2-ethanediamine (TPEN), a $Zn^{2+}$ chelator, showed a clear reduction of GZnP3 fluorescence. Both the quantum yield (QY) and the extinction coefficient of GZnP3 increased from the apo to $Zn^{2+}$-bound state (Supplementary Table 1), indicating that $Zn^{2+}$ causes increased brightness by positively affecting both photon absorption and fluorescent photon emission. GZnP3 has demonstrated a superior ability to quickly and sensitively detect in situ $Zn^{2+}$ fluctuations across the sub-nanomolar to nanomolar range (0.1–100 nM) with specific response to $Zn^{2+}$, thus it will be a vital tool for studying various intracellular $Zn^{2+}$ dynamics.

**TRPML1 channel is permeable to physiological levels of $Zn^{2+}$.**
Patch clamp studies have recorded inward current in cells expressing constitutively active mutant $TRPML1^{Va}$ on the plasma membrane evoked by 30 mM extracellular $Zn^{2+}$ [14], suggesting that TRPML1 might be permeable to $Zn^{2+}$. However, the physiological levels of $Zn^{2+}$ present inside the vesicular lumen are only in the hundreds of micromolar range[30]. Given that it is difficult to characterize the channel at micromolar concentrations of $Zn^{2+}$ via electrical current measurement, we utilized fluorescent $Zn^{2+}$ sensors to examine the permeability of TRPML1 channel to physiological $Zn^{2+}$ in live cells. We designed an experiment to record TRPML1-mediated $Zn^{2+}$ influx through the plasma membrane. TRPML1 is mostly localized on lysosomes and late endosomes, but previous studies have found that a small fraction of TRPML1 can localize on the plasma membrane in cells overexpressing TRPML1[31,32]. When HeLa cells expressing TRPML1 (also referred to as $TRPML1^{WT}$) were treated with synthetic agonist ML-SA1 (with similar potency as endogenous agonist phosphatidylinositol-3,5-bisphosphate)[33–35] in the presence of 100 μM $ZnCl_2$, we detected significantly quicker and higher $Zn^{2+}$ influx using the small molecule sensor FluoZin-3 (Fig. 2a & b). Since FluoZin-3 resides in multiple cellular compartments, we also recorded $Zn^{2+}$ influx into the cytosol by the GZnP3 sensor, which only resides in the cytosol and nucleus. Activation of TRPML1 accelerated cytosolic $Zn^{2+}$ increase (**blue plot**, Fig. 2c). In control cells without TRPML1 overexpression or expressing a TRPML1 dominant negative mutant ($TRPML1^{DDKK}$)[33], $Zn^{2+}$ was transported into the cytosol through a slower transport mechanism (**green and magenta plots**, Fig. 2c & d). Such slow $Zn^{2+}$ influx was barely detected by FluoZin-3 (Fig. 2a) because GZnP3 presents higher sensitivity than FluoZin-3 in response to sub-nanomolar changes in cytosolic $Zn^{2+}$. Compared to FluoZin-3, GZnP3 also demonstrated faster kinetics in response to TRPML1-mediated $Zn^{2+}$ influx. In addition, control treatments with ML-SA1 alone or $ZnCl_2$ alone showed no rapid increase in cytosolic $Zn^{2+}$ (Supplementary Fig. 3a, b). These data provide evidence that TRPML1 can transport physiological levels of $Zn^{2+}$.

**TRPML1 activation evokes cytosolic $Zn^{2+}$ signals in neurons.**
Next, we wanted to assess whether we could detect the release of $Zn^{2+}$ from intracellular compartments through the TRPML1 channel. TRPML1 loss of function causes MLIV disease with a prominent neurodegenerative phenotype, so we turned our focus to a neuronal cell model. Specifically, the hippocampus has been identified as a neuronal tissue with high levels of vesicular $Zn^{2+}$ [36] and hippocampal synaptic defects were observed in mouse models of MLIV[37]. As expected, in primary cultured rat hippocampal neurons co-expressing GZnP3 and mCherry-TRPML1, activation of TRPML1 with 50 μM ML-SA1 induced a significant increase in GZnP3 fluorescence, which can be quenched by the $Zn^{2+}$ chelator, TPEN (Fig. 3a, **blue plot** Fig. 3b & f). No signals were detected in neurons expressing $TRPML1^{DDKK}$ (**magenta plot** Fig. 3b & f). Additionally, pretreatment with 50 μM ML-SI4, a TRPML1 inhibitor[38], also blocked GZnP3 signals upon addition of ML-SA1 (Fig. 3c & f). Without overexpression of TRPML1, treatment with a more potent activator ML-SA5[38] evoked tiny and variable GZnP3 signals through endogenous TRPML1 channels (Fig. 3g), which is similar to the weak $Ca^{2+}$ release mediated by endogenous TRPML1[35]. In order to assess and compare the vesicular pool of $Zn^{2+}$ that can be liberated through TRPML1 channels, we used a commonly used TRPML1 overexpression system[16,17,33,35,38,39] that would allow rapid release of vesicular $Zn^{2+}$ upon TRPML1 activation for all the following experiments.

We performed additional experiments to confirm that the increase in GZnP3 fluorescence was indeed due to increased $Zn^{2+}$. Neurons pretreated with 100 μM TPEN showed no detectable increase in GZnP3 fluorescence upon addition of ML-SA1 (Fig. 3d & f). As TRPML1 is also permeable to $Fe^{2+}$, which has also been suggested in MLIV pathology[14], we further examined whether the GZnP3 signal was caused by a release of $Fe^{2+}$ to the cytosol. We found that TRPML1-induced GZnP3 signals were not affected by $Fe^{2+}$ chelator 2,2'-bipyridyl, though $Zn^{2+}$ chelation with TPEN rapidly quenched the signal (Fig. 3e & f). As TRPML1 activation can induce proton leak from acidic lysosomes[40], we examined whether the TRPML1-mediated $Zn^{2+}$ signals were underestimated due to the quenching of GZnP3 by reduced pH. We simultaneously recorded the changes in pH and GZnP3 signals and corrected the effects caused by pH (see Supplementary methods). The results found that the TRPML1-mediated GZnP3 signals were only slightly lower than the pH-corrected values (Supplementary Fig. 4 & 5).

Finally, we selected ZapCV2, a previously published genetically encoded zinc sensor, to test whether it could detect a TRPML1-mediated $Zn^{2+}$ release. Unlike GZnP3, ZapCV2 utilizes a different sensor design platform called Förster resonance energy transfer (FRET) to detect $Zn^{2+}$ with nanomolar binding affinity ($K_d = 2.3$ nM). ZapCV2 yields one of the highest in situ dynamic ranges (2.2) and maximal response above baseline (0.49), but this is substantially lower than the dynamic range and maximal response above baseline of GZnP3 (11 and 4, respectively) (Table 1). In neurons co-expressing mCherry-TRPML1, NES-ZapCV2 only detected an average maximum signal increase 6.2 ± 0.5% above baseline, which was significantly smaller than the 16.0 ± 2.4% increase of GZnP3 (Fig. 3h & i). Together, these data serve as the first direct evidence of TRPML1-mediated $Zn^{2+}$ release from intracellular stores to the cytosol in hippocampal neurons.

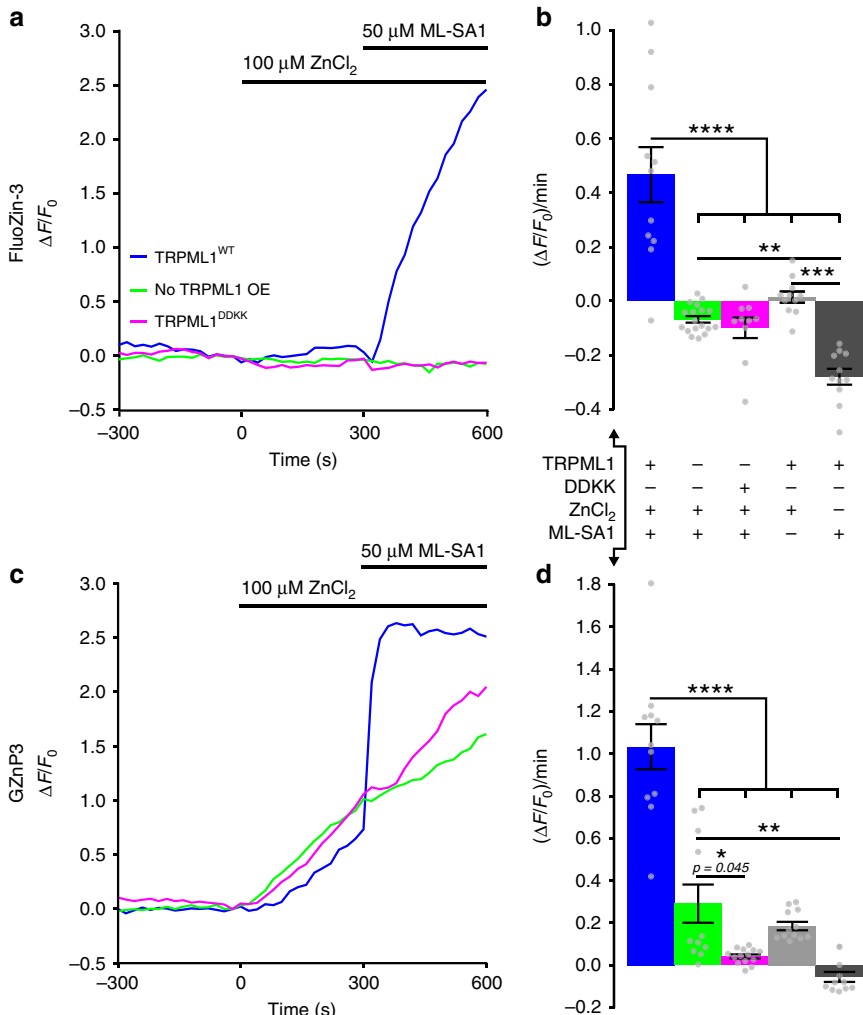

**Fig. 2** TRPML1 is permeable to physiological levels of $Zn^{2+}$. **a** Representative traces of FluoZin-3-stained HeLa cells either un-transfected (no TRPML1 OE, green), or expressing dominant negative (TRPML1$^{DDKK}$, magenta) or wild-type (TRPML1$^{WT}$, blue) mCherry-TRPML1. Cells were treated with 100 μM ZnCl$_2$ at 0 s then 50 μM ML-SA1 at 300 s. **b** Mean slope (±s.e.m.) of FluoZin-3 signal (($\Delta F/F_0$)/min) for 60 s following ML-SA1 addition for TRPML1$^{WT}$ (blue, $n = 11$ cells), no TRPML1 OE (green, $n = 18$), TRPML1$^{DDKK}$ (magenta, $n = 10$), TRPML1$^{WT}$ ZnCl$_2$ only (light gray, $n = 11$), TRPML1$^{WT}$ ML-SA1 only ($n = 11$). One-way ANOVA, with post-hoc Tukey HSD. **c** Representative traces of cells expressing GZnP3 alone (no TRPML1 OE, green), or co-transfected with dominant negative (TRPML1$^{DDKK}$, magenta) or wild-type (TRPML1$^{WT}$, blue) mCherry-TRPML1. Cells were treated as described in **a**. **d** Mean slope (±s.e.m.) of GZnP3 signal (($\Delta F/F_0$)/min) for 60 s following ML-SA1 addition for TRPML1$^{WT}$ (blue, $n = 11$ cells), no TRPML1 OE (green, $n = 11$), TRPML1$^{DDKK}$ (magenta, $n = 14$), TRPML1$^{WT}$ ZnCl$_2$ only (light gray, $n = 12$), TRPML1$^{WT}$ ML-SA1 only ($n = 11$). One-way ANOVA, with post-hoc Tukey HSD. ****$p < 0.0001$, ***$p < 0.001$, **$p < 0.01$, *$p < 0.05$. $p$-values displayed in italics above corresponding comparisons, where applicable. Source data are provided as a Source Data file

**Subcellular targeting of GZnP3 to detect local $Zn^{2+}$ signals.** After revealing this TRPML1-mediated $Zn^{2+}$ release in primary rat hippocampal neurons, we sought to identify in which subcellular compartment(s) this released $Zn^{2+}$ was stored. Though the endolysosomal localization of TRPML1 has been extensively studied in various cell types such as human skin fibroblasts[18], HeLa[19,20], and HEK293[14,21], we wanted to confirm this localization in hippocampal neurons overexpressing TRPML1 and investigate if TRPML1 is localized on synaptic vesicles which contain high concentration pools of $Zn^{2+}$.

Subcellular localization studies were carried out using primary cultured rat hippocampal neurons transfected with mCherry-TRPML1 and incorporated with markers of various subcellular compartments. TRPML1 colocalized most strongly with GFP-Rab7a, which decorates late endosomes and some lysosomes (Pearson's correlation coefficient, PCC = 0.810 ± 0.010) (Supplementary Fig. 6a, b). TRPML1 also strongly colocalized with the

acid-sensitive lysosomal dye, LysoTracker Green (Thermofisher) (PCC = 0.740 ± 0.026) (Supplementary Fig. 6a, b). However, mCherry-TRPML1 weakly localized with synaptophysin-EGFP (PCC = 0.455 ± 0.028) or FluoZin-3 (PCC = 0.561 ± 0.020) (Supplementary Fig. 6a, b). The synaptic vesicle localization was not statistically different from the negative control mitochondrial marker (mito-EGFP; PCC = 0.420 ± 0.059). These data confirm that overexpressed TRPML1 predominantly resides on endolysosomal membranes, not synaptic vesicles.

Next, we examined the distribution of $Zn^{2+}$ among different vesicles using a small molecule $Zn^{2+}$ sensor, FluoZin-3. Colocalization analysis of FluoZin-3 was performed against several vesicular markers in neurons. FluoZin-3 demonstrated comparable localization with both late endosomes (mCherry-Rab7a, Pearson's R value = 0.623 ± 0.021, n = 23) and synaptic vesicles (VAMP2-RFP, Pearson's R value = 0.678 ± 0.042, $n = 12$), suggesting that late endosomes might also make up a large

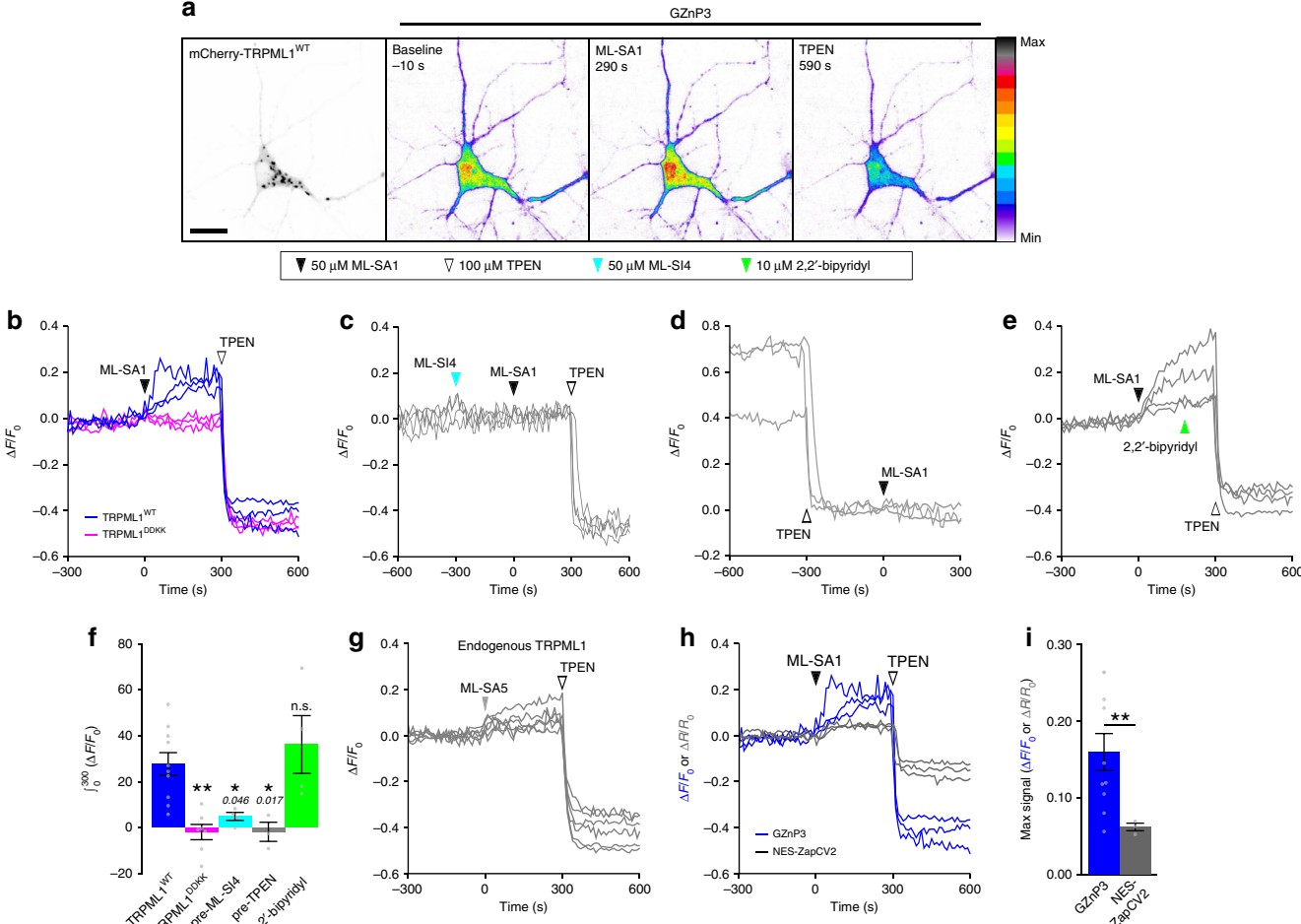

**Fig. 3** Activation of TRPML1 evokes cytosolic $Zn^{2+}$ signals in neurons. **a** Representative micrographs of primary rat hippocampal neurons co-expressing mCherry-TRPML1[WT] (left) and GZnP3 at baseline after ML-SA1 addition, or after TPEN treatment. Pseudocolor shows GZnP3 fluorescent intensity indicated by calibration bar (far right; minimum = white, maximum = black). Scale bar = 20 μm. **b** Representative traces of neurons co-expressing GZnP3 and mCherry-TRPML1[WT] (blue) or mCherry-TRPML1[DDKK] (magenta). Neurons were treated with ML-SA1 at 0 s and TPEN at 300 s. **c** Representative traces of neurons co-expressing GZnP3 and mCherry-TRPML1 pretreated with ML-SI4 at -300 s, ML-SA1 at 0 s, and TPEN at 300 s. **d** Representative traces of neurons co-expressing GZnP3 and mCherry-TRPML1 pretreated with TPEN at -300 s and ML-SA1 at 0 s. **e** Representative traces of neurons co-expressing GZnP3 and mCherry-TRPML1 treated with ML-SA1 at 0 sec, 2,2'-bipyridyl ($Fe^{2+}$ chelator) at 180 s, and TPEN at 300 s. **f** Mean integrated GZnP3 signal (±s.e.m.) within 300 s after ML-SA1 addition for TRPML1[WT] (blue, $n = 10$), TRPML1[DDKK] (magenta, $n = 7$), inhibitor pretreatment (pre-ML-SI4, cyan, $n = 4$), TPEN pretreatment (pre-TPEN, gray, $n = 3$), and $Fe^{2+}$ chelation (2,2'-bipyridyl, green, $n = 4$). One-way ANOVA with Dunnett's multiple comparison to TRPML1[WT]. **g** Representative traces of neurons co-expressing GZnP3 and NES-mCherry treated with 10–20 μM ML-SA5 at 0 s (gray arrow) and TPEN at 300 s. **h** Representative traces of neurons co-expressing mCherry-TRPML1 and either GZnP3 (blue) or NES-ZapCV2 (gray) treated as in **b**. **i** Mean maximum sensor signal (±s.e.m.) within 300 s after ML-SA1 addition for GZnP3 (blue, $n = 10$) and NES-ZapCV2 (gray, $n = 3$). Student's one-tailed $t$-test. **$p < 0.01$, *$p < 0.05$. n.s., not significant. $p$-values displayed in italics above corresponding comparisons where applicable. 50 μM ML-SA1 (black arrow), 100 μM TPEN (white arrow), 50 μM ML-SI4 (cyan arrow), 10 μM 2,2'-bipyridyl (green arrow). Source data are provided as a Source Data file

subset of high-$Zn^{2+}$ storage sites (Supplementary Fig. 6c). However, FluoZin-3 had poor colocalization with LysoTracker Red (Thermofisher) (Pearson's R value = 0.420 ± 0.037, $n = 8$). The data suggested that $Zn^{2+}$ is localized in both endolysosomal vesicles and synaptic vesicles, but also demonstrated non-specific localization of FluoZin-3 in various vesicles, which would limit its application to distinguish $Zn^{2+}$ among various vesicular pools. Instead, genetically encoded sensors like GZnP3 can be specifically targeted to organelles by fusion to various targeting signal peptides or proteins. Therefore, we targeted GZnP3 onto the surface of various subcellular vesicles to distinguish local TRPML1-mediated $Zn^{2+}$ release from different sources.

To measure $Zn^{2+}$ release directly from TRPML1 itself, we tagged GZnP3 to the cytosolic N-terminus of TRPML1 (Fig. 4a),

as previously performed with $Ca^{2+}$ sensor GCaMP3[35]. To measure lysosomal $Zn^{2+}$ release, we fused GZnP3 to the cytosolic C-terminus of LAMP1, an integral lysosomal protein (Fig. 4a). We also attached GZnP3 to the N-terminus of Rab7a for measurement of $Zn^{2+}$ release from late endosomes. Though our colocalizations suggested that TRPML1 has low colocalization with synaptic vesicles, we still wanted to examine if any ectopic localization of mCherry-TRPML1 on synaptic vesicles were responsible for the detected cytosolic $Zn^{2+}$ signal. To this end, we fused GZnP3 to the cytosolic C-terminus of synaptophysin (Fig. 4a) to measure local $Zn^{2+}$ release from synaptic vesicles. After making these new sensor constructs, we examined if the sensor response to $Zn^{2+}$ was affected by the fusion of the marker protein because fusion with a targeting motif can potentially alter

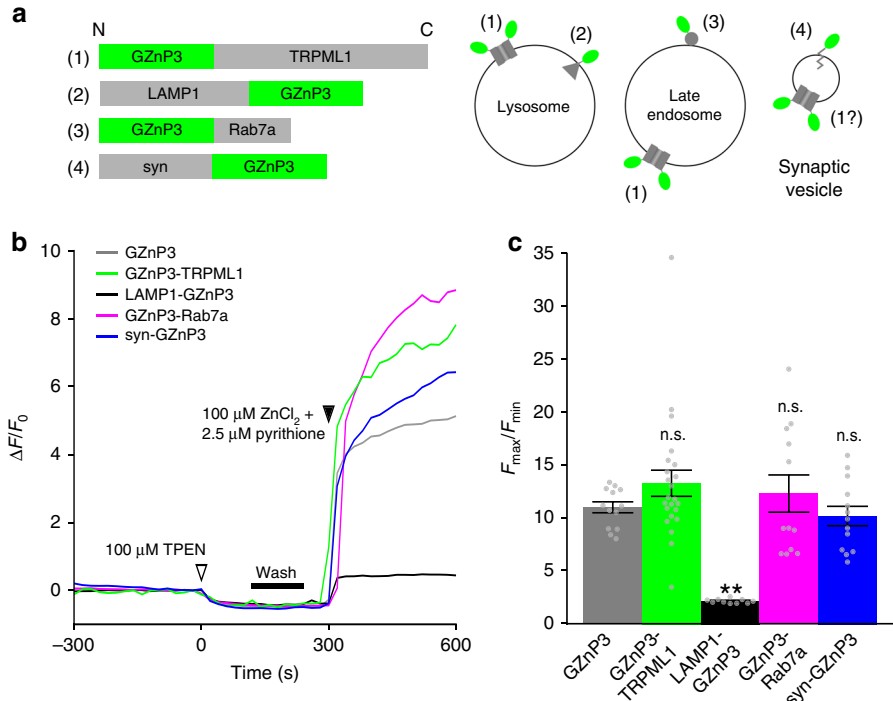

**Fig. 4** Target $Zn^{2+}$ sensor to subcellular compartments to detect local TRPML1-mediated signals. **a** Schematic of protein constructs for (1) GZnP3-TRPML1, (2) LAMP1-GZnP3, (3) GZnP3-Rab7a, and (4) synaptophysin-GZnP3 (syn-GZnP3) with N- and C-termini marked (left). Diagram of subcellular localizations of each targeted GZnP3 construct (right). **b** Representative traces of HeLa cells expressing GZnP3 (gray), GZnP3-TRPML1 (green), LAMP1-GZnP3 (black) GZnP3-Rab7a (magenta), or synaptophysin-GZnP3 (blue). Cells were treated with 100 μM TPEN at 0 s (white arrow), washed (black bar), then treated with 100 μM $ZnCl_2$ and 2.5 μM pyrithione at 300 s (black arrow). **c** Mean sensor dynamic range (±s.e.m.) of cytosolic control GZnP3 (gray, n = 13 cells) and vesicular-targeted variants: GZnP3-TRPML1 (green, n = 23), LAMP1-GZnP3 (black, n = 10), GZnP3-Rab7a (magenta, n = 12), and synaptophysin-GZnP3 (blue, n = 13). HeLa cells were treated as described in **b**. Bars indicate mean fold-change of sensor fluorescence ($F_{max}/F_{min}$). One-way ANOVA with Dunnett's multiple comparison to GZnP3. ****$p < 0.0001$, n.s. not significant. $p$-values listed in italics above corresponding comparisons where applicable. Source data are provided as a Source Data file

the tertiary structure of the sensor to perturb the sensor response. In HeLa cells, all vesicular-targeted sensors, except LAMP1-GZnP3, displayed dynamic ranges between approximately 10- and 13-fold, not significantly different from cytosolic GZnP3 (Fig. 4b, c). Therefore, GZnP3-TRPML1, GZnP3-Rab7a, and synaptophysin-GZnP3 could be used to sensitively detect $Zn^{2+}$ release from various vesicular compartments.

**TRPML1 releases $Zn^{2+}$ from endolysosomal vesicles**. The targeted variants of GZnP3 were then used to measure and compare local TRPML1-mediated $Zn^{2+}$ release from the three potential $Zn^{2+}$ storage sites: lysosomes, late endosomes, and synaptic vesicles. In primary hippocampal neurons, GZnP3-TRPML1 had punctate vesicular morphology similar to that of mCherry-TRPML1 (Supplementary Fig. 7a). GZnP3-Rab7a showed punctate morphology that visibly colocalized with mCherry-TRPML1, though it had a slight cytosolic signal (Supplementary Fig. 7b). Matching previous data in axon terminals, synaptophysin-GZnP3 puncta were in close proximity to, but did not visibly colocalize with mCherry-TRPML1 (Supplementary Fig. 7c).

Activation of TRPML1 induced a gradual increase in both cytosolic GZnP3 (Fig. 5a) and GZnP3-TRPML1 fluorescence (Fig. 5b, Supplementary Fig. 8a), which were not significantly different at either 30 seconds or 290 s after TRPML1 activation with ML-SA1. Interestingly, TRPML1 activation evoked rapid GZnP3-Rab7a signals (Fig. 5c, Supplementary Fig. 8b). The GZnP3-Rab7a signal was significantly higher than cytosolic GZnP3 30 seconds after ML-SA1 addition, but was statistically

lower than GZnP3 signal at 290 seconds. This suggests that GZnP3-Rab7a is able to detect local $Zn^{2+}$ release around late endosomes before it diffuses away to the cytosol, while cytosolic GZnP3 detects cytosolic $Zn^{2+}$ that are slowly equilibrated with $Zn^{2+}$ release from endolysosomes. The rapid signals (<20 s) detected by GZnP3-Rab7a suggest that activation of TRPML1 channels can generate endolysosomal $Zn^{2+}$ microdomains that are defined by localized high-$Zn^{2+}$ concentrations. To monitor $Zn^{2+}$ release from synaptic vesicles, we only focused on the axons and axon terminals that contain large amounts of synaptic vesicles[41] to avoid picking up signals from the endosomes (synaptophysin has slight endogenous localization on endosomes[42] and overexpression might enhance non-synaptic localization). Though synaptic vesicles colocalized strongly with FluoZin-3, treatment with ML-SA1 induced no significant synaptophysin-GZnP3 signal (Fig. 5d, Supplementary Fig. 8c), which suggests that the majority of TRPML1-mediated $Zn^{2+}$ signals are not released from synaptic vesicles.

To further distinguish late endosomes from lysosomes, we pretreated cells with 200 μM glycyl-L-phenylalanine 2-naphthylamide (GPN) for two minutes to disrupt lysosomes by osmotic lysis[43]. After GPN pretreatment, both GZnP3 and GZnP3-Rab7a detected TRPML1-mediated $Zn^{2+}$ release (Fig. 5f, g, Supplementary Fig. 9a, c). However, GZnP3-TRPML1 had little to no signal (Fig. 5f, g, Supplementary Fig. 9b), showing similar response as synaptophysin-GZnP3 trace without GPN treatment (Fig. 5d). The GZnP3-Rab7a and GZnP3-TRPML1 signal intensities both slowly decayed over ~5 min, however, possibly due to dilution of $Zn^{2+}$ throughout the cytosol, $Zn^{2+}$ reuptake

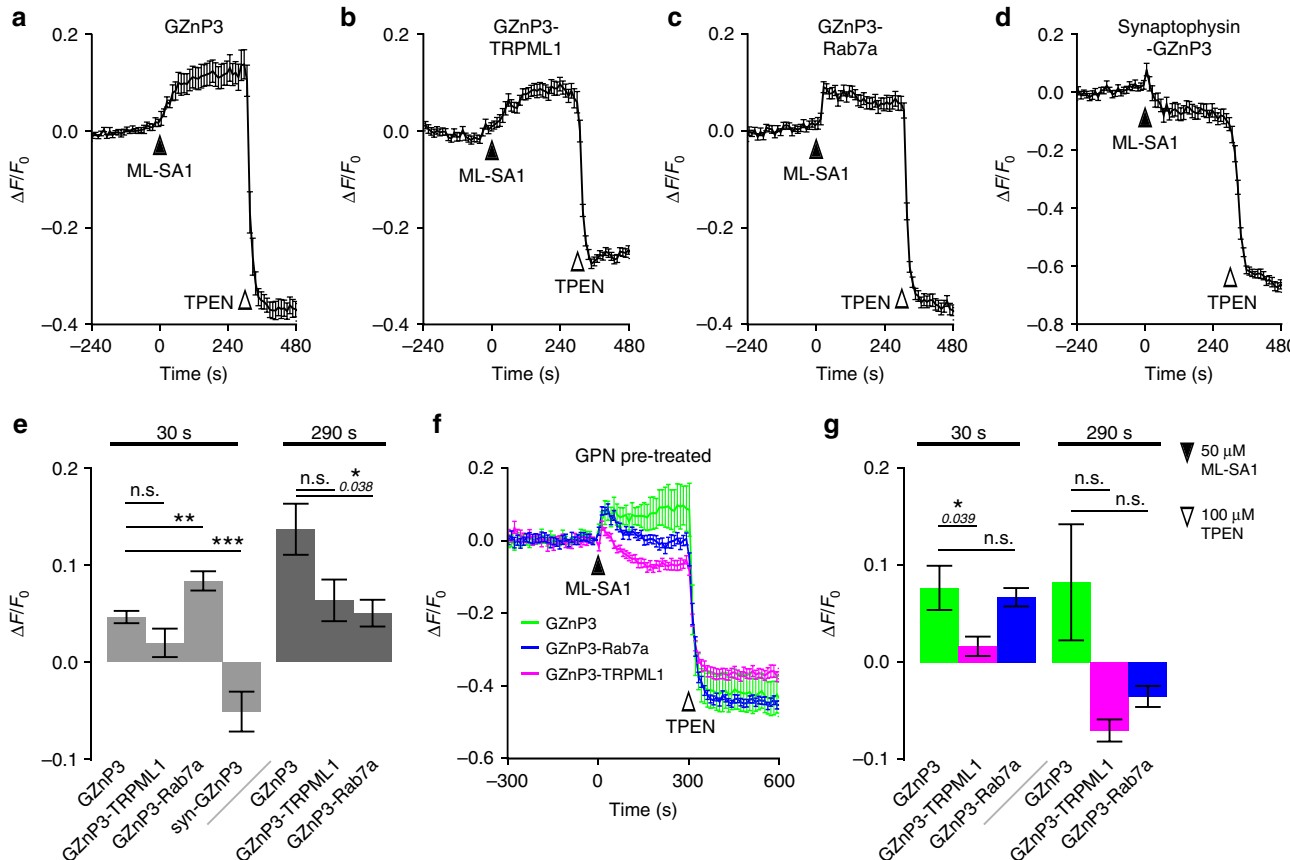

**Fig. 5** Activation of TRPML1 promotes $Zn^{2+}$ release from lysosomes and late endosomes, not synaptic vesicles. Average traces (±s.e.m.) of primary rat hippocampal neurons **a** coexpressing mCherry-TRPML1 and GZnP3, **b** expressing GZnP3-TRPML1, **c** coexpressing mCherry-TRPML1 and GZnP3-Rab7a, or **d** coexpressing mCherry-TRPML1 and synaptophysin-GZnP3, treated with 50 μM ML-SA1 at 0 s and 100 μM TPEN at 300 s. **e** Mean normalized GZnP3 signal (±s.e.m.) 30 and 290 s after ML-SA1 addition for cytosolic GZnP3 ($n = 10$ neuron soma), GZnP3-TRPML1 ($n = 48$ puncta from 5 neuron soma), GZnP3-Rab7a ($n = 62$ puncta from nine neuron soma), and synaptophysin-GZnP3 (syn-GZnP3; $n = 45$ puncta from three neurons). Two-tailed student's $t$-tests with Bonferroni correction for multiple comparisons. **f** Average traces (±s.e.m.) of primary rat hippocampal neurons coexpressing mCherry-TRPML1 and cytosolic GZnP3 (green), GZnP3-Rab7a (blue), or expressing GZnP3-TRPML1 (magenta), treated with 200 μM GPN from -600 to -480 seconds (not shown). **g** Mean normalized GZnP3 signal (±s.e.m.) 30 and 290 s after ML-SA1 addition for cytosolic GZnP3 ($n = 6$ neuron soma), GZnP3-TRPML1 ($n = 30$ puncta from four neuron soma), GZnP3-Rab7a ($n = 51$ puncta from six neuron soma). One-tailed student's $t$-tests with Bonferroni correction for multiple comparisons. 50 μM ML-SA1 (black arrow), 100 μM TPEN (white arrow). $***p < 0.001$, $**p < 0.01$, $*p < 0.05$, n.s. not significant. $p$-values listed in italics above corresponding comparisons where applicable. Source data are provided as a Source Data file

into vesicles, or membrane injury and clearance caused by lysosomal disruption. Together, our data suggest that GZnP3-Rab7a can monitor $Zn^{2+}$ release from late endosomes and TRPML1 activation can release $Zn^{2+}$ from both lysosomes and late endosomes.

**Comparison of TRPML1-mediated $Zn^{2+}$ and $Ca^{2+}$ in neurons.** Neurons are highly polarized with functionally different structures including the soma, axons, and dendrites. Recent work discovered that endolysosomal vesicles are not evenly distributed among the soma and neural processes. Rab7-positive late endosomes are found throughout the dendrites, while the mature and degradative lysosomes are mostly concentrated in the soma[44]. For this reason, we compared TRPML1-mediated $Zn^{2+}$ signal intensity between the soma and neurites. Interestingly, there were significantly higher TRPML1-mediated $Zn^{2+}$ signals in neurites than the soma detected by cytosolic GZnP3 (Fig. 6a, d), GZnP3-Rab7a (Fig. 6c, d), and GZnP3-TRPML1 (Fig. 6b, d).

In order to examine the possibility that the higher $Zn^{2+}$ signals in neurites are due to the spatially limited diffusion of $Zn^{2+}$ ions along these thin processes or different distributions of endolysosomal vesicles, we compared TRPML1-mediated $Ca^{2+}$ signals

between the soma and neurites by using GCaMP5, a sensitive green fluorescent calcium sensor extensively used in neuronal calcium studies[45]. Similar to GZnP3, GCaMP5 was fused to the N-terminus of both Rab7a and TRPML1 to measure $Ca^{2+}$ release from late endosomes and lysosomes. Cytosolic GCaMP5 showed a rapid spike in fluorescence upon addition of ML-SA1 and then immediate decay (Fig. 6e–g), consistent with previously reported TRPML1-mediated $Ca^{2+}$ signals in retinal pigment epithelium[46], COS-1[39], CHO[35], and HeLa[39] cells. Different from $Zn^{2+}$ signals, the TRPML1-mediated $Ca^{2+}$ signals were significantly lower in neurites as compared to the soma as detected by GCaMP5 (Fig. 6e, h) and GCaMP5-Rab7a (Fig. 6g, h). Conversely, GCaMP5-TRPML1 showed no significant difference between soma and neurites (Fig. 6f, h). Such data suggest that the endolysosomes, especially late endosomes store higher levels of $Zn^{2+}$, and less $Ca^{2+}$ in neurites than in the soma.

To further verify that the higher $Zn^{2+}$ signals in neurites were due to more $Zn^{2+}$ released from the Rab7a-positive late endosomes in neurites than the soma, we compared the TRPML1-mediated $Zn^{2+}$ signal between neurites and soma after lysosomal $Zn^{2+}$ was depleted in neurites with 200 μM GPN. GZnP3 showed slightly higher signals in neurites than the soma,

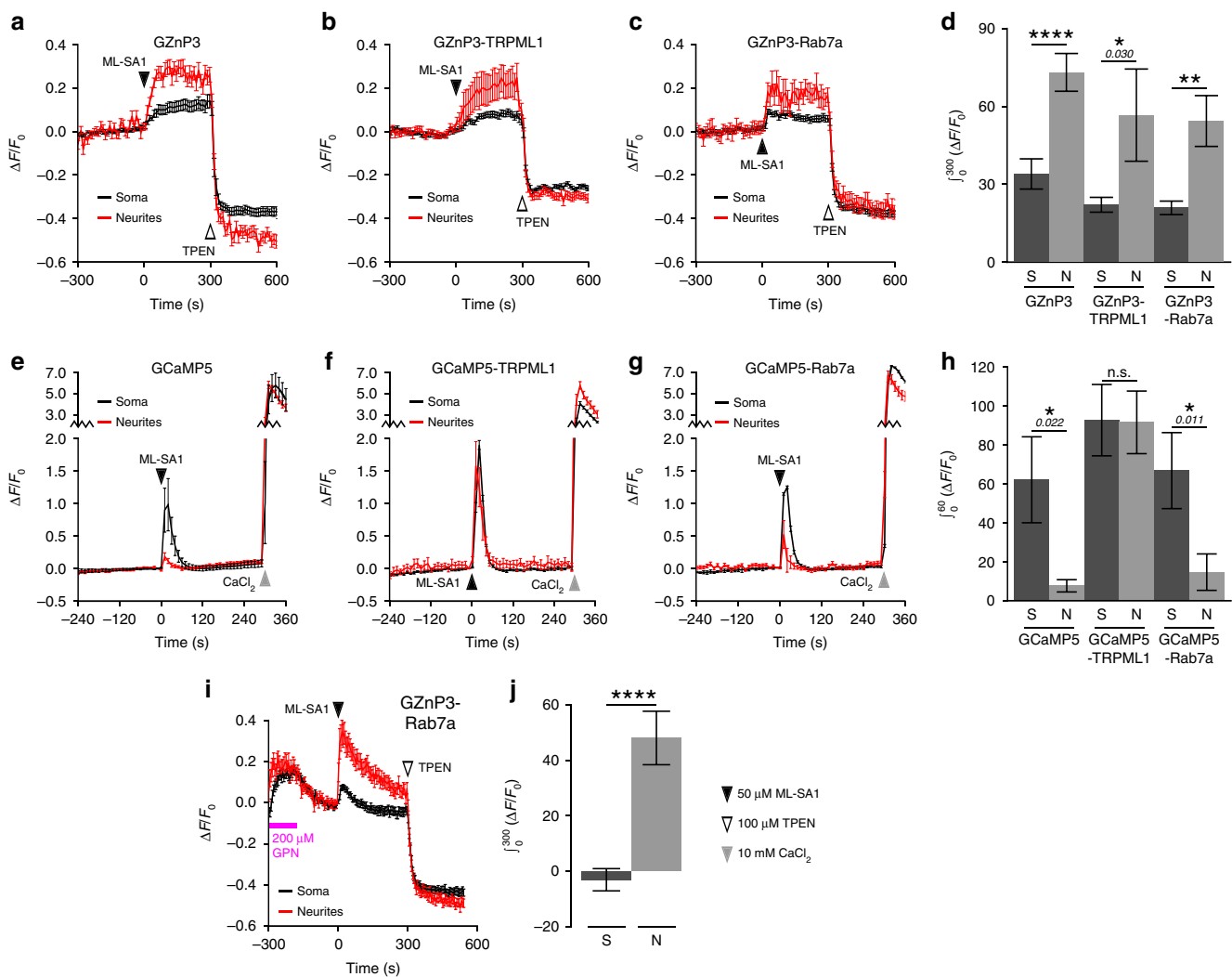

**Fig. 6** TRPML1 mediates greater $Zn^{2+}$ release and less $Ca^{2+}$ release in neurites than in the soma. Average traces (±s.e.m.) from soma (black) or neurites (red) of primary rat hippocampal neurons co-expressing **a** mCherry-TRPML1 and GZnP3, **b** expressing GZnP3-TRPML1, or **c** co-expressing mCherry-TRPML1 and GZnP3-Rab7a. **d** Mean integrated GZnP3 signal (±s.e.m.) 300 s after ML-SA1 addition between soma (dark) and neurites (light) for GZnP3 (soma, $n = 10$; neurites, $n = 70$ puncta), GZnP3-TRPML1 (soma, $n = 48$ puncta; neurites, $n = 54$ puncta), and GZnP3-Rab7a (soma, $n = 62$ puncta; neurites, $n = 28$ puncta). One-tailed Student's t-tests. Average traces (±s.e.m.) from soma (black) or neurites (red) of primary rat hippocampal neurons **e** co-expressing mCherry-TRPML1 and GCaMP5, **f** expressing GCaMP5-TRPML1, or **g** co-expressing GCaMP5-Rab7a and mCherry-TRPML1. **h** Mean integrated GCaMP5 signal (±s.e.m.) 60 s after ML-SA1 addition between soma (dark) and neurites (light) for GCaMP5 (soma, $n = 8$; neurites, $n = 68$ puncta), GCaMP5-TRPML1 (soma, $n = 30$ puncta; neurites, $n = 66$ puncta), and GCaMP5-Rab7a (soma, $n = 25$ puncta; neurites, $n = 42$ puncta). One-tailed Student's t-tests. **i** Average traces (±s.e.m.) from soma (black) and neurites (red) of primary rat hippocampal neurons co-expressing GZnP3-Rab7a and mCherry-TRPML1 pretreated with 200 μM GPN (magenta bar). **j** Mean integrated GZnP3-Rab7a signal (±s.e.m.) 300 s after ML-SA1 addition to neurons pretreated with GPN between soma (dark) and neurites (light) (soma, n = 41 puncta; neurites, n = 59 puncta). 50 μM ML-SA1 (black arrow), 100 μM TPEN (white arrow), 10 mM CaCl₂ (gray arrow). One-tailed Student's t-test. ****$p < 0.0001$, **$p < 0.01$, *$p < 0.05$, n.s. not significant. p-values listed in italics above corresponding comparisons where applicable. S, soma. N, neurites. Source data are provided as a Source Data file

while GZnP3-TRPML1 had no observable signal in either soma or neurites after lysosomal disruption (Supplementary Fig. 10a, b). As expected, after GPN treatment, GZnP3-Rab7a detects significantly higher $Zn^{2+}$ release from late endosomes in neurites than the soma (Fig. 6i, j), indicating that $Zn^{2+}$ in the Rab7a-positive late endosomes are higher in neurites than in the soma. Overall, our data suggest that the Rab7a-positive late endosomes store higher $Zn^{2+}$ and less $Ca^{2+}$ in neurites than the soma.

Interestingly, it was also observed that unlike the $Ca^{2+}$ signals, the $Zn^{2+}$ signals did not return quickly to baseline after ML-SA1 treatment. Such difference was not caused by variations in sensor kinetics because GZnP3 showed comparable or even faster turn-on and turn-off responses compared with GCaMP5 (Supplementary

Fig. 11, see Supplementary Methods). Sustained $Zn^{2+}$ signals might be due to slower buffering mechanisms than $Ca^{2+}$. To test this, we utilized a depolarization condition to accumulate high $Ca^{2+}$ and $Zn^{2+}$ in the cytosol[47]. After depolarization was removed, $Ca^{2+}$ was reduced by 99%, while $Zn^{2+}$ was only buffered 12% within 15 min (Supplementary Fig. 12, see Supplementary Methods). Therefore, the cytosolic $Zn^{2+}$ signals, either released from vesicular compartments or induced by influx, can last longer than $Ca^{2+}$.

**Cell heterogeneity of TRPML1-mediated $Zn^{2+}$ and $Ca^{2+}$ signals.** TRPML1 has nearly ubiquitous tissue expression[48], though TRPML1 loss of function in MLIV patients presents a

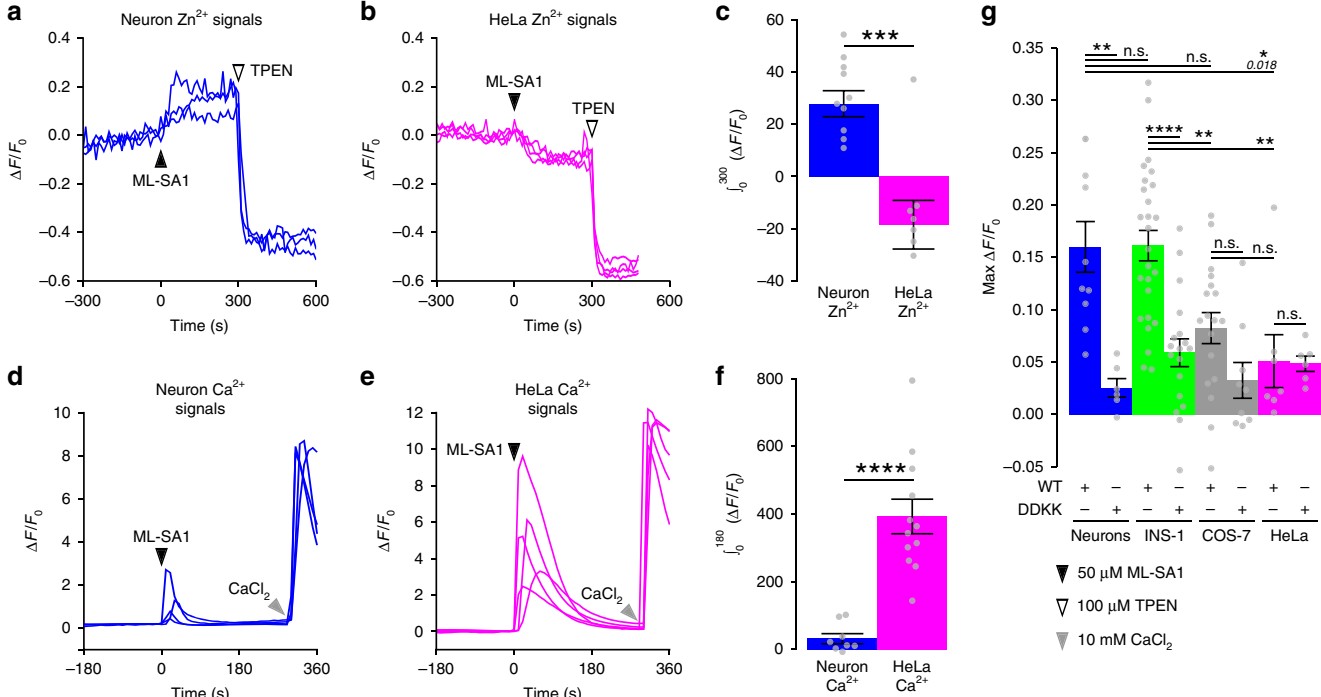

**Fig. 7** TRPML1-mediated $Zn^{2+}$ signals are higher in neurons and INS-1 cells than other mammalian cell types. **a** Representative traces of primary rat hippocampal neurons co-expressing GZnP3 and mCherry-TRPML1. Neurons were treated with 50 μM ML-SA1 at 0 s (black arrow) and 100 μM TPEN at 300 s (white arrow). **b** Representative traces of HeLa cells co-expressing GZnP3 and mCherry-TRPML1. Cells were treated as described in **a**. **c** Mean integrated GZnP3 signal (±s.e.m.) 300 s after ML-SA1 addition for both neurons (blue, $n = 10$ cells) and HeLa cells (magenta, $n = 7$). Two-tailed Student's $t$-test. **d** Representative traces of primary rat hippocampal neurons co-expressing GCaMP5 and mCherry-TRPML1. Neurons were treated with 50 μM ML-SA1 at 0 s (black arrow) and 10 mM $CaCl_2$ at 300 s (gray arrow). **e** Representative traces of HeLa cells co-expressing GCaMP5 and mCherry-TRPML1. Cells were treated as described in **d**. **f** Mean integrated GCaMP5 signal (±s.e.m.) 180 s after ML-SA1 addition for both neurons (blue, $n = 8$) and HeLa cells (magenta, $n = 12$). Two-tailed Student's $t$-test. **g** Mean GZnP3 signal maximum (±s.e.m.) 300 s after ML-SA1 addition for both $TRPML1^{WT}$ and $TRPML1^{DDKK}$ in neurons (blue; WT, $n = 10$; DDKK, $n = 7$), INS-1 cells (green; WT, $n = 26$; DDKK, $n = 18$), COS-7 cells (gray; WT, $n = 18$; DDKK, $n = 9$), and HeLa cells (magenta; WT, $n = 7$; DDKK, $n = 6$). One-way ANOVA with post-hoc Tukey HSD. ****$p < 0.0001$, ***$p < 0.001$, **$p < 0.01$, *$p < 0.05$, n.s. not significant. $p$-values listed in italics above corresponding comparisons where applicable. Source data are provided as a Source Data file

strong neurodegenerative phenotype. Therefore, we compared TRPML1-mediated $Zn^{2+}$ signals among different cell types. In contrast to neurons (Fig. 7a), HeLa cells co-expressing TRPML1-mCherry and GZnP3 showed a decrease in GZnP3 signal upon activation of TRPML1 (Fig. 7b, c), likely due to decreased pH. HeLa cells showed a statistically lower $Zn^{2+}$ release than neurons (Fig. 7c), indicating the endolysosomal vesicles may not store high $Zn^{2+}$ in HeLa cells. We also sought to analyze whether TRPML1-mediated $Ca^{2+}$ release was different in HeLa cells as compared to neurons. HeLa cells co-expressing GCaMP5 and mCherry-$TRPML1^{WT}$ were treated with 50 μM ML-SA1. HeLa cells showed much larger $Ca^{2+}$ release upon TRPML1 activation, which decayed much slower than in neurons (Fig. 7e). Unlike $Zn^{2+}$ signals, the TRPML1-mediated $Ca^{2+}$ signals were significantly higher in HeLa cells than in neurons (Fig. 7d–f).

Next, we sought to investigate whether the TRPML1-mediated $Zn^{2+}$ release varied across other commonly used mammalian cell models such as the pancreatic beta-cell line (INS-1) and African green monkey kidney fibroblast-like cell line (COS-7). Interestingly, there was no significant difference between TRPML1-mediated $Zn^{2+}$ signals in INS-1 as compared to neurons (Fig. 7g). Similar to neurons, INS-1 cells expressing mCherry-TRPML1 also showed a significantly larger TRPML1-mediated $Zn^{2+}$ signal than INS-1 cells expressing $TRPML1^{DDKK}$ (Fig. 7g). Similar to HeLa cells, $Zn^{2+}$ release was not significantly higher in COS-7 cells expressing mCherry-$TRPML1^{WT}$ as compared to $TRPML1^{DDKK}$ (Fig. 7g). The data suggest that the neurons and pancreatic beta

cells contain high $Zn^{2+}$ in the endolysosomal vesicles that can be liberated through the TRPML1 channel, while HeLa cells and COS-7 cells lack such endolysosomal $Zn^{2+}$ pools.

## Discussion

Though $Zn^{2+}$ has been unequivocally identified as a vital metal ion for cellular health, the dynamics of labile $Zn^{2+}$ have not been well characterized due to its low intracellular concentration and a lack of sensitive probes that are able to detect its physiological fluctuations in the sub-nanomolar to nanomolar range. While a growing toolbox of fluorescent $Ca^{2+}$ sensors have been established, which vary in their sensitivity and fluorescent spectra, there are limited tools available to study $Zn^{2+}$ signaling dynamics in live cells.

The development of GZnP3 has filled such a gap for the scientific community giving an unprecedented ability to study cellular $Zn^{2+}$ dynamics with sub-nanomolar sensitivity in real time. GZnP3 binds labile $Zn^{2+}$ with a $K_d$ of 1.3 nM, giving it the ability to detect this metal ion in the sub-nanomolar range. Though other genetically encoded $Zn^{2+}$ probes have similar binding affinities (Table 1), GZnP3 has approximately an 11-fold dynamic range from its apo state to $Zn^{2+}$ saturation (17-fold in vitro), making it the most sensitive protein-based $Zn^{2+}$ sensor currently available for monitoring sub-nanomolar cellular $Zn^{2+}$ dynamics, between 100 pM and 1 nM. It has high specificity for $Zn^{2+}$ over a range of other biologically relevant cations, including $Ca^{2+}$ and $Fe^{2+}$. Together, these characteristics permit the use of GZnP3 to

observe minute changes in cytosolic $[Zn^{2+}]$ from the high pico-molar to low nanomolar range. Though it is a powerful tool, it does have limitations for its use in the dynamic cellular environment. Primarily, like many GFP-based fluorophores, we are aware of its sensitivity to changes in pH. To account for this, however, we have established a normalization method to obtain pH-corrected GZnP3 signals by simultaneously recording GZnP3 $(Zn^{2+})$ and pHuji[49] (pH) signals (Supplementary Fig. 5).

Here, we provided the first direct evidence that $Zn^{2+}$ can be released from intracellular compartments to the cytosol and showed that TRPML1 channels can mediate such $Zn^{2+}$ release in neurons. One of the major challenges of current biological studies of cellular $Zn^{2+}$ signals is to distinguish $Zn^{2+}$ among various vesicular compartments. In this work, we created sensors that allow us to record and compare $Zn^{2+}$ release from lysosomes, late endosomes, and synaptic vesicles. These constructs revealed several important findings. Our results support the presence of localized $Zn^{2+}$ signals in microdomains near endolysosomes before equilibrium is reached. Additionally, GZnP3-TRPML1 and GZnP3-Rab7a can detect different pools of $Zn^{2+}$ release. GZnP3-TRPML1 tends to detect $Zn^{2+}$ release from lysosomes because it failed to detect $Zn^{2+}$ signals when lysosomes were disrupted by GPN. Instead, GZnP3-Rab7a can still detect $Zn^{2+}$ release from non-lysosomal vesicles after lysosomal disruption (Fig. 5f, g). Although high pools of $Zn^{2+}$ are present in the synaptic vesicles[50] (Supplementary Fig. 6c), synaptophysin-GZnP3 close to the presynaptic vesicles at axon terminals failed to detect significant $Zn^{2+}$ signals when TRPML1 channels are activated (Fig. 5d, e). In addition, TRPML1 channel has weak colocalization with synaptic vesicles (Supplementary Fig. 6a, b). Overall, we conclude that the majority of TRPML1-mediated $Zn^{2+}$ release is from endolyso-somal compartments.

Interestingly, we found that TRPML1-mediated $Zn^{2+}$ signals are especially higher in neurites than the soma and the majority of such higher $Zn^{2+}$ was released from Rab7a-positive late endosomes. Biochemical evidence revealed that $Zn^{2+}$ can inhibit cathepsin (lysosomal protease) activity at concentrations as low as 100 nM[51], thus our findings integrate well into recent work about the polarized distributions of endolysosomal vesicles and degradative activity in neurons:[52] mature lysosomes with high degradative activity are localized within the soma of neurons, while non-degradative Rab7a-positive late endosomes are distributed along the dendrites[44]. The negative correlation between endolysosomal $Zn^{2+}$ and lysosomal degradative function in the neuronal soma and neurites implicates the potential involvement of $Zn^{2+}$ in lysosomal degradative activity and MLIV pathology.

Additionally, there are stark differences between the signal of TRPML1-mediated $Zn^{2+}$ release versus TRPML1-mediated $Ca^{2+}$. First, TRPML1-mediated $Ca^{2+}$ release in neurons shows a rapid spike and almost immediate decay, while TRPML1-mediated $Zn^{2+}$ release is relatively slower than $Ca^{2+}$ and has prolonged elevation. Such differences might be caused by different metal ion buffering mechanisms. There are several cellular mechanisms that regulate $Ca^{2+}$ reuptake into various compartments including the mitochondrial $Ca^{2+}$ uniporter (MCU) and ER $Ca^{2+}$ ATPase (SERCA), or export from the cell through Plasma Membrane $Ca^{2+}$ ATPase (PMCA)[53], ensuring the fast recovery of $Ca^{2+}$ back to baseline concentrations. In nearly all cell types, $Ca^{2+}$ has multiple downstream signaling targets, and these sequestration mechanisms can prevent cytotoxicity from prolonged elevation of ectopic cytosolic $Ca^{2+}$[54]. In contrast, $Zn^{2+}$ is buffered by relatively slower mechanisms including $Zn^{2+}$ buffering proteins[55,56], such as metallothionein and $Zn^{2+}$ export transporters (ZnT family)[6,8]. Such slow mechanisms might allow the small amount of released $Zn^{2+}$ signals to effectively interact with $Zn^{2+}$ sensing proteins such as metal transcription factor MTF1[57], tyrosine

phosphatase[58], and other signaling molecules to regulate neuron function. Second, there is an inverse correlation between TRPML1-mediated $Zn^{2+}$ and $Ca^{2+}$. TRPML1-mediated $Zn^{2+}$ signals are higher in neurons than HeLa cells, while TRPML1-mediated $Ca^{2+}$ signals are lower in neurons than HeLa cells (Fig. 7). In addition, when TRPML1 is activated in neurons, $Zn^{2+}$ signals are higher in neurites than soma, while $Ca^{2+}$ signals are lower in neurites than soma. This negative correlation suggests that increase in $Zn^{2+}$ in the endolysosomal vesicles might be complemented with a decrease in $Ca^{2+}$. The increase of $Zn^{2+}/Ca^{2+}$ ratio in the endolysosomal vesicles in neurons, especially in the neurites, might be associated with unique roles of these vesicles in neuron function.

In summary, we reported the creation of new $Zn^{2+}$ probes GZnP3, GZnP3-TRPML1, and GZnP3-Rab7a for detection of sub-nanomolar $Zn^{2+}$ dynamics. With these tools, we provided the first direct evidence that TRPML1 can mediate $Zn^{2+}$ release from endolysosomal vesicles to the cytosol. Such $Zn^{2+}$ signals demonstrate cell specificity and are especially higher in neurites than the soma in neurons. Overall, our work provides instrumental tools to investigate the biological function of endolysosomal $Zn^{2+}$ pools and implicates TRPML1-mediated $Zn^{2+}$ signals in proper neuronal function.

## Methods

**Animals.** Pregnant Sprague Dawley rats were purchased from Charles River (strain: 400). Animal treatment and maintenance were performed by the University of Denver Animal Facility (AAALAC accredited). All experimental procedures using animals were approved by the Institutional Animal Care and Use Committee (IACUC) of the University of Denver.

**In vitro characterization of $K_d$, metal specificity.** For sensor protein expression, the sensor was cloned into the pBAD vector and expressed in Top10 *Escherichia coli* upon addition of 0.2% arabinose. Sensor protein was purified by $Ni^{2+}$ ion affinity chromatography using Ni-NTA chelating sepharose beads, and the 6X Histidine tag was removed with a homemade TEV protease treated in 20 mM Tris, 100 mM NaCl, and a pH 8 buffer at RT overnight (sensor to TEV protease ratio was 10:1). The sensor protein was stored at 4 °C in the dark for up to one week from the day of purification. $Zn^{2+}$ titrations were performed in HEPES buffer (150 mM HEPES, 100 mM NaCl, 0.5 mM TCEP and 10% glycerol, pH 7.4) with 2 μM sensor protein and $Zn^{2+}$ buffering solutions with defined $Zn^{2+}$ concentrations[5]. The sensor was reduced with TCEP for 10 min prior to performing analysis. Purified sensor protein was titrated with $Zn^{2+}$ to determine the fluorescence intensity as a function of $Zn^{2+}$ concentration. The sensor response to $Zn^{2+}$ was reported as an apparent dissociation constant ($K_d'$), which reflects the $Zn^{2+}$ concentration midway between the $F_{min}$ and $F_{max}$. Metal selectivity was measured using 2 μM purified sensor protein in HEPES buffer[29]. The sensors were treated with 12 μM EDTA followed by 50 μM of each metal ($ZnCl_2$, $CaCl_2$, $MgCl_2$, $CuCl_2$, $MnCl_2$, $CoCl_2$, $NiSO_4$, $FeCl_2$, KCl, $FeCl_3$, $AlCl_3$, $CrCl_3$). $FeCl_2$ was maintained in the reduced oxidation state ($Fe^{2+}$) using ascorbic acid as a reducing agent to prevent oxidation to $Fe^{3+}$[59]. The in vitro fluorescence measurements used for $Zn^{2+}$ titrations and metal specificity studies were made on a Tecan fluorescence plate reader using the following parameters: excitation, 488 nm; emission, 515 nm; emission bandwidth, 5 nm.

**Biophysical characterization of GZnP3.** All biophysical characterizations were performed on purified sensor in 30 mM MOPS (pH 7.4), 100 mM KCl, and 0.5 mM TCEP. The quantum yield was determined using fluorescein as a reference, diluted in 0.1 M NaOH[29], over a dilution series between 0.01 to 0.05 AU absorbance at 494 nm using a Varian Cary 100 Bio UV-Vis spectrophotometer. Sensor proteins were incubated with 10 μM $ZnCl_2$ or 15 μM TPA ($Zn^{2+}$ chelator) to obtain the quantum yield of the sensor in the bound and apo state, respectively. The same dilution series was then run through a Cary Eclipse fluorescence spectrophotometer fluorimeter with excitation at 488 nm and emission from 500 nm to 600 nm using 1 nm slits, 1 second integration time, and 1 nm steps. The total fluorescence intensities were obtained by integrating the emission spectrum of each sample (500–600 nm) and plotted against the absorbance reading at the maximum wavelength (494 nm). The quantum yield was obtained from the slopes of each line and by plugging into the equation $\Phi_{Sensor} = \Phi_{Standard} \times$ (Slope of sensor/Slope of standard); $\Phi_{Standard} = 0.925$ for fluorescein. Extinction coefficients were calculated from the absorbance spectrum for a set concentration of purified sensor in the presence of 15 μM TPA and 10 μM $ZnCl_2$. The sensor protein concentrations were determined using A446 nm after alkali denaturation with 0.1 M NaOH for 3 min to eliminate fluorescence and generate an absorbance for $\varepsilon_{446\ nm}$. The extinction

coefficient was calculated using the equation $\varepsilon_{sensor}/44{,}000 =$ Absorbance of sensor/Absorbance in NaOH.

**DNA constructs.** Plasmids were constructed by molecular cloning and verified by sequencing. Site directed mutagenesis was used to introduce mutations to designed positions on GZnP1[29]. GZnP3 was fused to Rab7a amplified from mCherry-Rab7a (Addgene #55127) and TRPML1 from TRPML1-HA (Addgene #18825)[21]. GZnP3 was fused to LAMP1 amplified from LAMP1-RFP (Addgene #1817)[60]. GZnP3 was fused to synaptophysin from synaptophysin-GCaMP3 (a kind gift from Dr. Susan M Voglmaier, UCSF). GZnP1, GZnP2, GZnP3, and pHuji (amplified from TfR-pHuji, Addgene #61505)[49] were amplified and cloned into the pDisplay vector.

**Non-neuronal cell culture and transfection.** HeLa cells were maintained in high glucose Dulbecco's Modified Eagle Medium (DMEM) with 10% fetal bovine serum (FBS) at 37 °C, 5% $CO_2$. INS-1 cells were maintained in Roswell Park Memorial Institute (RPMI) 1640 Medium supplemented with 10 mM HEPES, 5 mM sodium pyruvate, 50 μM 2-mercaptoethanol, and 10% FBS. PEI (polyethyleneimine) transfection reagent was used for the transfection in non-neuronal cells (dissolved to 1 mg/ml in water at pH 7.2 with long-term storage at −20 °C, short term storage at 4 °C). Transfection was performed when the cells were ~40–50% confluent. For each transfection reaction in non-neuronal cells, 2–3 μL of PEI transfection reagent and 1–1.25 μg of DNA were mixed in 250 μL Opti-MEM. The mixture was incubated for a minimum of 25 min at room temperature before direct addition to one imaging dish containing the cells.

**Primary rat hippocampal neuron culture and transfection.** Primary hippocampal neurons were prepared from rat embryos at embryonic day 18 (E18) in dissociation medium containing 10X HBSS, 1 M HEPES buffer (pH 7.3) and 50 μg/mL gentamycin. The hippocampi were minced and treated with 1000 U/mL papain added to the dissection medium, and dissociated by trituration in 1 mg/mL DNase I. Cells were plated on 1 mg/mL poly-L-lysine-coated 14 mm round glass coverslips at a density of 40,000 cells/coverslip per dish in neuron plating medium (MEM supplemented with glucose and 5% FBS). After checking that cells were adhered, neuron plating medium was replaced with Neurobasal medium (Thermofisher) supplemented with 0.3X GlutaMAX (Thermofisher) and 1X B-27 (Thermofisher). Cultures were maintained at 37 °C, 5% $CO_2$. Neurons were transfected between DIV 6-15, using the Lipofectamine 3000 transfection kit (Thermofisher) in 500 μL Opti-MEM. The reagent-DNA mixture was incubated for a minimum of 25 min at room temperature before direct addition to the neuron imaging dishes. Before adding reagent-DNA, 1 mL media was removed from each imaging dish and syringe filtered with an equal volume of fresh neuron culture media (50:50 media). After incubation at 37 °C for 4 h, neurons were washed three times with 1 mL prewarmed neuron culture media. Then, 2 mL of the 50:50 media was added to the neuron imaging dishes and they were incubated at 37 °C until imaging.

**Live cell microscopy.** Cells were imaged 48 h post-transfection, and cells were washed three times with the indicated imaging buffer immediately before imaging. All imaging was performed on an inverted Nikon/Solamere CSUX1 spinning disc confocal microscope with a ×40 (for time-lapse) or ×60 (for colocalization) 1.4 NA oil immersion objective. Data were collected using MicroManager software and analyzed with Fiji (ImageJ).

**Colocalization.** Colocalizations with mCherry-TRPML1: Between DIV 6-15, primary cultured rat hippocampal neurons were either transfected only with mCherry-TRPML1, or co-transfected with a late endosomal marker (GFP-Rab7a) or a synaptic vesicle marker (synaptophysin-EGFP). After 48 h, neurons that were only transfected with mCherry-TRPML1 were loaded with either 1 μM Lyso-Tracker Green (Thermofisher) or 2 μM FluoZin-3 (Thermofisher). To determine the c.

Colocalizations with FluoZin-3: Between DIV 6-15, primary cultured rat hippocampal neurons were either transfected with mCherry-TRPML1[WT], mCherry-TRPML1[DDKK], mCherry-Rab7a, or VAMP2-RFP. After 48 h, neurons were loaded with 2 μM FluoZin-3, and neurons that were un-transfected were simultaneously loaded with 1 μM LysoTracker Red (Thermofisher).

All neurons were then washed and imaged in $Zn^{2+}$-free buffer. Colocalizations were performed using the Coloc 2 plugin for ImageJ. Point spread functions for the ×60 objective were measured for both 488/525 nm (full width at half maximum, FWHM = 2.58 pixels, PSF 1st σ = 154.5 nm) and 561/605 nm (FWHM = 2.75 pixels, PSF 1st σ = 164.9 nm) excitation/emission wavelengths and an average FWHM of 2.663 pixels was used as the input for the Coloc 2 plugin. To determine the Pearson's coefficient for random localization or autocorrelation, the colocalization was analyzed between TRPML1-mcherry and mito-EGFP that have no biological association as a negative control.

**In situ metal specificity.** For in situ $Ca^{2+}$ specificity, HeLa cells were transfected with GZnP3 and imaged in phosphate-free HHBSS (containing 1.26 mM $Ca^{2+}$). After collecting a 5-min baseline, 10 μM ionomycin (in DMSO) was added to

rapidly influx $Ca^{2+}$ into the cytosol. After 5 min, 100 μM TPEN was added to the cells to chelate all $Zn^{2+}$. For in situ $Fe^{2+}$ specificity, HeLa cells were transfected with GZnP3 and imaged in 0 $Ca^{2+}$ phosphate-free HHBSS. After collecting a 5-min baseline, 10 μM 2,2'-bipyridyl (in DMSO) was added to chelate cellular $Fe^{2+}$. After 5 min, 100 μM TPEN was added to the cells to chelate all $Zn^{2+}$.

**pDisplay sensor sensitivity assay.** HeLa cells were pretreated with 100 μM TPEN to bring sensor fluorescence to its minimum, then cells were washed and imaged with buffer containing 500 μM TCEP to prevent sensor oxidation. Starting at the proposed resting cytosolic concentration, we tested the sensors' response across a thousand-fold range of physiologically relevant concentrations: 100 pM, 1 nM, 10 nM, and 100 nM. Standard solutions of various $Zn^{2+}$ concentrations ($Zn^{2+}$/Chelator-buffered solutions) were prepared at defined concentrations[5]. Images were acquired every 10 s, with a 200 ms exposure of 488 nm laser excitation at 10 mW power. A stable baseline was collected for 120 s before the addition of $Zn^{2+}$.

**TRPML1 activation in neurons.** Primary cultured rat hippocampal neurons were transfected between 6 and 15 days in vitro (DIV). After 48 h, neurons were washed and imaged in 0 $Ca^{2+}$, 0 $Zn^{2+}$ HHBSS. After collecting a baseline signal for 5 min, neurons were treated with 50 μM ML-SA1 (or 10–20 μM ML-SA5 where indicated) to open TRPML1 channels. For inhibition experiments, after baseline was collected, 50 μM ML-SI4 was added for five minutes before ML-SA1 addition. As an internal control to confirm the presence of functional sensors, 100 μM TPEN was added to quench intracellular $Zn^{2+}$ for cells expressing GZnP3, or 10 mM $CaCl_2$ was added to influx $Ca^{2+}$ for cells expressing GCaMP5. For GZnP3 and GCaMP5 variants, images were acquired every 5–10 seconds, with a 200 ms exposure of 488 nm laser excitation at 10 mW power. For ZapCV2, images were acquired every 10 seconds with 200 ms exposures of 445 nm laser excitation at 10 mW power, capturing emissions at both 480 nm (CFP) and 535 nm (FRET). The background corrected FRET and CFP intensities were used to calculate the FRET ratio (R).

**Testing Dynamic Range of vesicular-targeted GZnP3 variants.** HeLa cells were imaged in 0 $Zn^{2+}$, phosphate-free HHBSS buffer, 48 h post-transfection. After a 5 min baseline, cells were treated with 100 μM TPEN to collect the minimum sensor fluorescence ($F_{min}$). TPEN was then removed by washing the cells three times with 0 $Zn^{2+}$ buffer. Finally, 100 μM $ZnCl_2$ and 2.5 μM pyrithione (a $Zn^{2+}$ ionophore) were added simultaneously to influx $Zn^{2+}$ and generate maximum sensor fluorescence ($F_{max}$). $F_{max}$ was then divided by $F_{min}$, representing the sensor's dynamic range of intensity as a fold-change relative to its minimum fluorescence. Images were acquired every 20 s, with a 200 ms exposure of 488 nm laser excitation at 10 mW power.

**Data Analysis.** Imaging data were analyzed with Fiji (ImageJ) and raw data output from Fiji were analyzed using Excel in combination with JMP software (JMP®, Version 13.0. SAS Institute Inc., Cary, NC). Statistical analysis were performed using Excel or JMP software, either performing unpaired t-tests or one-way ANOVA with appropriate post-hoc corrections. All t-tests were preceded by an F-test to analyze variance equality. t-tests were then performed with consideration for the determined variance equality or inequality. ANOVA with comparisons to a single control used Dunnett's post-hoc correction, and ANOVA with comparisons across all groups used post-hoc Tukey HSD. All measurements were taken from distinct samples, though multiple regions of interest (ROIs) were selected from a single neuron for measuring distinct puncta or various unique structures. No ROI was measured repeatedly. When selecting a ROI in the soma for cytosolic sensors that were also present in the nucleus, care was taken to avoid selecting nuclear areas. For cells loaded with FluoZin-3, care was taken to avoid high intensity puncta likely representing vesicles with high-$Zn^{2+}$ or concentrated dye. For time traces, background fluorescence was subtracted from ROIs. For single wavelength sensors, changes in fluorescent intensity ($\Delta F = F - F_0$) were normalized to the baseline preceding the 0 second time point ($F_0$), indicated as $\Delta F/F_0$. For FRET sensors, images were acquired for both the acceptor and donor emission wavelengths, and the changes in the ratio of acceptor to donor emission was calculated for each time point ($\Delta R = R - R_0$). Again, these signals were normalized to the baseline preceding the 0 s time point ($R_0$), indicated as $\Delta R/R_0$.

**Reporting summary.** Further information on research design is available in the Nature Research Reporting Summary linked to this article.

## Data availability
Original data are available from the corresponding author upon reasonable request. Source data underlying Figs. 1–7 and Supplementary Figures 1-6, 10-12 are provided as a Source Data file.

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

## Acknowledgements

We would like to acknowledge the following sources for general financial support: University of Denver startup fund, NIH Grant R00EB017289 (to Y.Q.), NIH Grant R01NS110590 (to Y.Q.), and R01NS062792 (to H.X.).

## Author contributions

T.F.M contributed design and generation of vesicular-targeted sensors, live cell imaging, experimental design, figure generation, data analysis and interpretation, and wrote the manuscript. C.Z. contributed rat hippocampal neuron primary culture preparation and maintenance, live cell imaging, and in situ sensor screening with the pDisplay assay. D.H.F. contributed sensor development, live cell imaging, and in vitro sensor characterization. A.M.D. contributed construction of vesicular-targeted sensors, live cell imaging, and data analysis. K.D.L. contributed sensor development and assisted in vitro work. H.X. contributed experimental design and data interpretation. Y.Q. supervised the project and contributed experimental design, data interpretation, and wrote the manuscript.

## Competing interests

The authors declare no competing interests.
