## [Transparent Peer Review File · Nature Communications]

Reviewers' comments:

Reviewer #1 (Remarks to the Author):

The manuscript describes a new genetically encoded sensor GZnP3 which is able to sense zinc at sub-nanomolar concentrations. Additional experiments also demonstrate that GZnP3 can be used to monitor the dynamic release of zinc from neuronal endolysosomal vesicles through the TRPML1 channel. Given that genetic mutations in TRPML1 lead to the autosomal recessive lysosomal storage disease MLIV, important findings from this work include that TRPML1 can facilitate the release of zinc from intracellular compartments. Another particularly novel aspect of the work is the tethering of the GZnP3 sensor to the TRPML1 channel and other endo/lysosomal markers. By tethering the sensor to these proteins the authors are able to monitor the localized release of zinc from a specific compartment into a smaller 'microdomain' within the cytosol.

Overall this is a well controlled study which provides evidence that the TRPML1 channel can release zinc ions. I have the following questions and comments.

1) While data is presented to show that zinc can be released through the TRPML1 channel, it remains unclear if this release of zinc has any physiological significance. There are a wide range of proteins whose activity is known to be dependent upon zinc (e.g. increased cytosolic zinc will increase the activity of MTF1- and Metallothionein 1 gene expression). Does the addition of ML-SA1 (or knockdown of TRPML1) alter the activity of endogenous zinc proteins? Also is anything known about the natural conditions which lead to cation release via the TRPML channel?

2) The data shows that ML-SA1 addition leads to the rapid release of zinc and not calcium from neurons and the rapid release of calcium and not zinc from HeLa cells. Presumably if this channel is able to release multiple divalents metal ions there would be competition between them. Do the differences between the cell types simply reflect the levels of Ca^{2+} and Zn^{2+} in each compartment under a given growth conditions. As an example, if neurons were pre-exposed to higher levels of calcium before the addition of the ML-SA1 agonist, would this lead to the loss of zinc release (because now calcium would be the most abundant cation in the compartment and would therefore outcompete zinc)

3) The results suggest that there are significant differences in the TRPML-1 mediated release of calcium and zinc (calcium being rapid, and zinc being released over a longer period). Is it possible that these are a result of differences in the sensors and not the cellular environment? e.g. could it be differences in the rate at which each of the respective cations are exchanged from the sensor?

Minor comments

1) The TRPML - DDKK control is critical as it provides evidence that zinc release is via the TRPML channel. Given the importance of this control, evidence should be presented to show that this mutant accumulates within cells at similar level to TRPML (e.g. an immunoblot comparing the levels of the two sensors).

Amanda Bird

Reviewer #2 (Remarks to the Author):

In the present manuscript, Minckley and co-authors present a novel zinc reporter several characteristics of which appear to be superior to some of the currently used probes. As many

recent zinc probes, GZnP3 is genetically encoded, which may be used to target it to desired compartments, and its dynamic range is broader than its predecessors' ones. While there are benefits of the GZnP3 's increased dynamic range, their discussion is presented in very broad strokes. Specifically, the dye does not appear to be more sensitive below 1 nM and the significance of the observed effects between 1 and 10 nM is not obvious. It does not change the impact of the dye, but a more nuanced discussion would have been useful.

Based on the prior evidence (from the co-author's lab) of zinc fluxes through the lysosomal ion channel TRPML1, the authors fuse GZnP3 to proteins resident in the lysosomes and test zinc fluxes in response to TRPML1 activation in several cell types. Similar to the prior evidence, TRPML1 activation by the pharmacological mean released zinc from the lysosome. Some fusion constructs were more effective than others, but the source or meaning of these differences are not entirely clear. Further, the authors show that the lysosomes in cultured neurons (or the cells themselves?) handle zinc differently than do cultured HeLa cells. This is not entirely surprising considering how different these cell types are with regards to the expression levels of zinc transporters and many other aspects of physiology.

The discussion of zinc role and physiology is somewhat bombastic. The authors claim that their work suggests that zinc sequestration or release could be important for endolysosomal maturation into degradative lysosomes; there is no evidence for this, or such evidence should be better discussed. The discussion of the buffering rate of calcium vs zinc is not supported by numbers. The inverse relation between zinc and calcium handling can be explained by differences in transporter levels, which may be completely cell-type specific or even reflect the different course of adaptation to cell culture conditions.

In summary, this is a novel iteration of a dye that convincingly and quantitatively reports zinc. The studies are technically sophisticated and the dye is clearly useful. However, the biological novelty of the studies presented here can be improved or explained better.

Reviewer #3 (Remarks to the Author):

Qin etc. reports a genetically encoded sensor, GZnP3, which demonstrates unprecedented sensitivity for Zn²⁺ at sub-nanomolar concentrations. Using GZnP3 as well as GZnP3-derived vesicular targeted probes, we provide the first direct evidence that Zn²⁺ can be released from endolysosomal vesicles to the cytosol in primary hippocampal neurons through the TRPML1 channel. I would like to suggest its publication in the Nat. Comm. after correction of the following major points:

1. In the introduction part, the authors add the following references. i.e.

i) A Two-Input Fluorescent Logic Gate for Glutamate and Zinc. C Yin, F Huo, NP Cooley, D Spencer, K Bartholomew, CL Barnes. Timothy E. Glass. ACS chemical neuroscience, 2017, 8 (6), 1159-1162.

ii) Three-Input Logic Gates with Potential Applications for Neuronal Imaging. Kenneth S. Hettie, Jessica L. Klockow, and Timothy E. Glass. J. Am. Chem. Soc., 2014, 136 (13), 4877–4880.

iii) A specific fluorescent probe for zinc ion based on thymolphthalein and its application in living cells. F Huo, J Kang, Y Zhang, C Yin. Sensors and Actuators B: Chemical. 2018, 262, 263-269.

iv) Development of fluorescent zinc chemosensors based on various fluorophores and their applications in zinc recognition. J Li, C Yin, F Huo. Dyes and Pigments. 2016, 131, 100-133.

2. What was the effect of Na⁺、K⁺、Fe³⁺、Fe²⁺、Cr³⁺ etc. life elements?

3. What is the effect of zinc citrate, zinc gluconate, etc?

4. Please recheck the English throughout word by word before publishing.

Reviewer #4 (Remarks to the Author):

SUMMARY

Minckley and colleagues are reporting an improved GFP-based genetically encoded zinc indicator (GZnP3) and its application for understanding the role of the channel TRPML1 in mediating neuronal Zn²⁺ dynamics. The results are potentially impactful for two reasons. First, as an indicator with relatively high affinity and response amplitude, GZnP3 may fill a sensitivity gap in the toolbox of zinc indicators; as such, it may prove to be of utility to an expanding community investigating the role of zinc dynamics in neurons and other cell types. Second, they have used their novel tool to get insights into TRPML1, a channel involved in the disease Mucopolysaccharidosis type IV disease (MLIV); while previous studies have focused on TRPML1's role in calcium signaling, this study is focusing on zinc signaling. The manuscript is overall well researched, written, and illustrated. The major concerns to address are: (1) the lack of comparisons between GZnP3 and published protein-based zinc indicators of other families, (2) difficulties in interpreting the GZnP3's responses given its sensitivity to calcium, and (3) difficulties in interpreting the experiments with the tagged GZnP3 and GPN treatment. Should these concerns be fully addressed, we believe this work would be of broad interest to this journal's readership.

MAJOR CONCERNS

1. The relative usefulness of GZnP3 versus other indicators has not been clearly established

The authors mention in the introduction that other sensors "have high binding affinity in the picomolar range, they are limited with very low dynamic range (<4)". The overall dynamic range (no zinc to saturating zinc) is of low relevance for use of these indicators for physiological measurements given that all cells have a basal Zinc concentration and saturating zinc concentrations are unlikely to be encountered. More critical is the overall response amplitude within the range of signals investigated. Given this, it would be reasonable to expect a comparison of GZnP3 and a published indicator that has the largest dynamic range around the expected change of Zinc in neurons (100 pM – 1 nM according to the authors). These indicators could be compared in neurons in response to ML-SA1, showing both average traces (like Figure 3b but just for the wild-type TRPML1) and quantification with bar graphs as done elsewhere.

2. Can calcium dynamics be complicating the interpretation of the results?

Figure 1d suggests that calcium can reduce the baseline fluorescence of the GZnP3 by ~50%. Since TRPML1 can conduct calcium, can this impact the interpretation of all experiments in this paper? For example, might the response of GZnP3 in the soma be smaller than that in neurites (Figure 6a) solely because calcium fluxes in the soma are much larger than in the neurites (as per Figure 6e)? Also, note that Figure 1d has the only bar graph in the paper where statistical comparisons were not provided.

3. Experiments in Figures 5 and 6 require further clarification

The authors mentioned that "After GPN treatment, TRPML1-mediated GZnP3 signals were still increased but to a slightly reduced level compared with the condition without GPN pretreatment". But in Figure 5d, the authors show that this change is not statistically significant. Given this, I would suggest the authors just mention that they did not observe a statistically significant change in GZnP3 signals following GPN pretreatment (or that they observed a slight reduction, but that this change did not reach statistical significance). Moreover, addition of ML-SA1 may not have been performed at steady state as GZnP3 signals were still decreasing when ML-SA1 was added. As a result, the response to ML-SA1 in the +GPN treatment could have been underestimated.

In Figure 5b, the traces are also not at steady state when ML-SA1 is being added. This may be because some GZnP3-TRPML1 is on lysosomal membranes that are being lysed. These effects

complicate the analysis of this experiment.

In Figure 5d, the bar graph suggests that, following GPN treatment, ML-SA1 didn't induce fluorescence responses to Zinc. However, the average trace (5c) clearly shows a response. I think the problem is that the red trace wasn't at steady state when ML-SA1 was added. The short response to ML-SA1 is thus counterbalanced by the sloping baseline. Another concern is that since the GZnP3 domain in GZnP3-Rab7a is cytoplasmic, we would expect that its dF/F at time 300 sec (just before TPEN application) would be similar to that of GZnP3 (which is also cytosolic) in the +GPN condition (as well as in the -GPN condition).

The authors suggest there are no signals from synaptophysin-GZnP3 but there is a clear signal in Figure 5e. However, it's not clear if it's a real signal or an artifact due to application of the agonist.

Providing representative pictures of neurons in the various conditions (with GZnP3, GZnP3-TRPML1, and GZnP3-Rab7a) would be useful. In cases where GPN treatment was used, please provide pictures before treatment and at around time 0, before ML-SA1 was added.

The authors analyze "fluorescent puncta". Do they get similar results if they include the whole soma?

Finally, why is the response of GZnP3-Rab7a to ML-SA1 initially larger in the +GPN compared with the -GPN condition? See Figures 6c and i.

3. Please discuss the potential caveats of using an TRPML1 overexpression system and its impact on the main results reported in this paper

The 3rd paragraph of the discussion claims that this work demonstrates the measurement of *physiological* fluctuations in Zn^{2+} . Because TRPML1 was overexpressed and because an agonist was used, I don't think this claim can be made.

MINOR COMMENTS

In Figure 1b, is the response of GZnP3 delayed compared with the application of the zinc buffers?

In the first subsection of the Results section, Supplementary Figure 1a should be Supplementary Figure 1b and vice-versa (or better, switch the figures in the Supplementary info).

In Figure 2b, why is gray bar negative? Is this a pH effect?

Figure 3a, the response would be best represented as a change from the baseline i.e. we would suggest not to show the baseline image, but re-color the ML-SA1 and TPEN image to represent the % change from the baseline.

In the Data Analysis section, it is written that "All t-tests took variance equality into consideration". Please clarify. What was done if the samples did not show equal variance? Or did all samples have equal variance?

Response to Reviewers

We appreciate all the valuable comments and constructive suggestions. The manuscript has been significantly revised in the ways suggested by all reviewers. In summary, we have made the following major changes:

1. To address one major issue raised by both reviewer 2 and reviewer 4 regarding the sensor sensitivity for Zn^{2+} between 100 pM – 1 nM, we have included a new **Table 1** to compare GZnP3 with all previously published genetically encoded Zn^{2+} sensors. In addition, we made a further comparison between GZnP3 with a FRET based Zn^{2+} sensor ZapCV2 to validate and compare TRPML1-mediated Zn^{2+} signals (**Fig. 3h & i**).
2. We completed *in vitro* analysis of additional metals: Fe^{2+} , K^+ , Fe^{3+} , Al^{3+} , Cr^{3+} (**Fig. 1d**).
3. For comparison between GZnP3, GZnP3-Rab7a, and GZnP3-TRPML1 with pre-GPN treatment, we performed new experiments in which sensor intensity was allowed to sit at a stable baseline for 5 minutes before TRPML1 activation by ML-SA1 (**Fig. 5f**).
4. We have revised **Fig. 5** to separate the sensor traces under different conditions for each sensor for the sake of better clarification.
5. We analyzed and compared GZnP3 (Zn^{2+} sensor) and GCaMP5 (Ca^{2+} sensor) *in situ* kinetics (**Supplementary Fig. 11**).
6. We compared buffering of Ca^{2+} and Zn^{2+} in neurons after cells were loaded with high Ca^{2+} or Zn^{2+} induced by depolarization (**Supplementary Fig. 12**).

In order to comply with the Nature Communications Reporting Summary, we have added relevant *p* values to figures with statistical comparisons and reported the statistical methods used for each comparison in the figure legends. The statistics section of Methods was also updated to report the measurement of distinct samples.

Due to word limit constraints, we have reduced background information and maintained a concise discussion. The detailed point by point responses to each of the reviewer's comments are included below:

Reviewer 1:

Comment 1: “While data is presented to show that zinc can be released through the TRPML1 channel, it remains unclear if this release of zinc has any physiological significance. There are a wide range of proteins whose activity is known to be dependent upon zinc (e.g. increased cytosolic zinc will increase the activity of MTF1- and Metallothionein 1 gene expression). Does the addition of ML-SA1 (or knockdown of TRPML1) alter the activity of endogenous zinc proteins?”

Response: We thank the reviewer for this suggestion. TRPML1-mediated Zn^{2+} signals can potentially impact neuron function by interacting with Zn^{2+} sensing proteins. Increased cytosolic Zn^{2+} can be sensed by the metal transcription factor MTF1 to regulate neuronal gene expressions (Pfaender et al., *Neural Plast* 2016). In addition, picomolar dynamic changes of Zn^{2+} can modulate the activities of tyrosine phosphatase (Wilson & Maret, *J Biol Chem* 2012), which is an important enzyme for hippocampal synapse formation and learning (Fuentes et al., *PloS One* 2012). Zn^{2+} can also modulate the brain-derived neurotrophic factor (BDNF) signaling which is involved in neuronal differentiation, synaptogenesis and synaptic plasticity (Zagrebelsky et al., *Neuropharmacology* 2013, Scharfman & MacLusky, *Neuropharmacology* 2014, Leal et al., *Brain research* 2015). TRPML1 knockdown was found to alter cellular Zn^{2+} homeostasis and both lysosomal and cytosolic Zn^{2+} levels were elevated (Kukic et al., *Biochem*

J 2013). In addition, the Metallothionein 2a mRNA levels are higher in TRPML1 knockdown cells than wild type cells when exposed to high Zn^{2+} (Kukic et al., *Biochem J* 2013). So far, we are only beginning to understand the roles of intracellular Zn^{2+} signals in cells and most of results derive from biochemical evidence or studies on whole cells that are loaded with excess Zn^{2+} . These methods are not sufficient to uncover the physiological significance of transient bursts of TRPML1-mediated Zn^{2+} signals in live cells. The endogenous agonist of TRPML1 is the transient endolysosome-specific phosphoinositide PI(3,5)P₂ (Dong, et al., *Nat Commun* 2010), therefore, TRPML1-mediated Zn^{2+} signals are small, transient and local within microdomain regions nearby endolysosomes so that they might not cause global effects on whole cells. Our future work will utilize the new sensors created here to track the dynamic changes of TRPML1-mediated Zn^{2+} signals and reveal its roles in neuron function.

Due to word limits, we only included one sentence to discuss the potential interactions of TRPML1-mediated Zn^{2+} signals with Zn^{2+} sensing proteins (pg. 11, last paragraph).

Comment 2: “Also is anything known about the natural conditions which lead to cation release via the TRPML channel?”

Response: The known endogenous agonist of TRPML1 is the signaling lipid, phosphatidylinositol-3,5-bisphosphate (PI(3,5)P₂) (Dong et al, *Nat Commun* 2010; Fine et al, *Nat Commun* 2018). PI(3,5)P₂ lipids are extremely scarce as compared to other phosphatidylinositol species, and are mainly confined to late endosomes and lysosomes. PI(3,5)P₂ is generated from the phosphorylation of phosphatidylinositol-3-phosphate (PI3P) by the kinase, PIKfyve, which is associated with the Vac14 scaffold protein. PI(3,5)P₂ binds to the TRPML1 channel within the plane of the lipid bilayer, and the 3' phosphate causes the S4-S5 linker to pull away from the channel pore, therefore dilating the lower gate of the pore allowing ions to flow through (Chen et al, *Nature* 2017).

Due to word limits, we only included a brief description of PI(3,5)P₂ in the results (pg. 4) to discuss the endogenous agonist of TRPML1 channel as compared to the synthetic agonist ML-SA1.

Comment 3: “The data shows that ML-SA1 addition leads to the rapid release of zinc and not calcium from neurons and the rapid release of calcium and not zinc from HeLa cells. Presumably if this channel is able to release multiple divalents metal ions there would be competition between them. Do the differences between the cell types simply reflect the levels of Ca²⁺ and Zn²⁺ in each compartment under a given growth conditions. As an example, if neurons were pre-exposed to higher levels of calcium before the addition of the ML-SA1 agonist, would this lead to the loss of zinc release (because now calcium would be the most abundant cation in the compartment and would therefore outcompete zinc)”

Response: We agree with the reviewer that our results might be caused by the different levels of cations within the endolysosomal vesicles. Unfortunately, it is not known how cells accumulate Ca²⁺ into lysosomes. High intracellular Ca²⁺ in neurons can be quickly taken up by mitochondrial Ca²⁺ uniporter and ER Ca²⁺ ATPase (SERCA), as well as exported from the cell through Plasma Membrane Ca²⁺ ATPase (PMCA) (see review Bagur & Hajnoczky, *Mol Cell* 2017) and unregulated Ca²⁺ overload can trigger severe neuron death (see review Orrenius & Nicotera, *J Neural Transm Suppl* 1994). As such, it is impossible to only load Ca²⁺ into lysosomes without perturbing the whole cell Ca²⁺ homeostasis and neuron health.

Comment 4: “The results suggest that there are significant differences in the TRPML-1 mediated release of calcium and zinc (calcium being rapid, and zinc being released over a

longer period). Is it possible that these are a result of differences in the sensors and not the cellular environment? e.g. could it be differences in the rate at which each of the respective cations are exchanged from the sensor?"

Response: We appreciate the constructive advice. We have tested the kinetics of the GCaMP5 and GZnP3 sensors *in situ* in order to determine whether or not GZnP3 has slower kinetics than GCaMP5 (**Supplemental Fig. 11**). In HeLa cells expressing cytosolic GZnP3 or GCaMP5, there was no significant difference between the turn on rate of these sensors upon addition of 10 μM Zn^{2+} and 2.5 μM pyrithione, or 1 mM Ca^{2+} and 5 μM ionomycin, respectively. GZnP3 had a statistically faster turn off rate than GCaMP5 in response to cation reduction by chelators or intracellular sequestration/buffering. This suggests that the Zn^{2+} sensor GZnP3 has binding kinetics that are similar or even faster than Ca^{2+} sensor GCaMP5.

Comment 5: "The TRPML - DDKK control is critical as it provides evidence that zinc release is via the TRPML channel. Given the importance of this control, evidence should be presented to show that this mutant accumulates within cells at similar level to TRPML (e.g. an immunoblot comparing the levels of the two sensors)."

Response: Though we agree this would be an ideal way to measure and compare expression of TRPML1^{WT} and TRPML1^{DDKK}, there are significant limitations to performing an immunoblot for our experimental system. The transfection efficiency for primary neuron cultures is extremely low, with a roughly estimated maximum 15-20% of all neurons transfected at our earliest transfection date (DIV5). Alternatively, we have quantified the expression of mCherry-TRPML1^{WT} and mCherry-TRPML1^{DDKK} by analyzing the average fluorescent intensity of mCherry in neuron soma. Since mCherry is fused with the TRPML1 channel, the fluorescence of mCherry can indirectly represent the TRPML1 channel expression level. There was no statistical difference between the intensity of mCherry-TRPML1^{WT} and mCherry-TRPML1^{DDKK} (see figure below), suggesting similar expression and accumulation of mutant and wildtype TRPML1.

Measurements were taken from confocal micrographs collected using 10 mW (561 nm) laser intensity, 100 ms exposure time, and a setting of "60" for EMGAIN. Observable nuclei (devoid of clear mCherry fluorescence) were excluded from the soma regions of interest (ROIs) given the variable nuclear size caused by cell to cell variation and Z-axis positioning of the confocal imaging plane. To correct for any other variables that could have affected fluorescent intensity between neurons and imaging dishes, average intensity of mCherry within the soma was

normalized to the background fluorescence of each image ($\frac{\text{soma-background}}{\text{background}}$). Background ROIs were selected to have approximately similar size to the soma ROI and covered areas devoid of any observable cell debris or aberrant fluorescence.

In addition to our data suggesting similar expression as WT in neurons, TRPML1-DDKK has also been used (and confirmed) extensively as a negative control in previously published research (Cao et al, *J Biol Chem* 2017; Sun et al, *Autophagy* 2018; Shen et al, *Nat Commun* 2012).

Reviewer 2:

Comment 1: “the dye does not appear to be more sensitive below 1 nM and the significance of the observed effects between 1 and 10 nM is not obvious.” “a more nuanced discussion would have been useful.”

Response: We thank the reviewer for the suggestions. In order to highlight that GZnP3 presented improved sensitivity Zn²⁺ dynamics ranging from 100 pM to 1 nM Zn²⁺, we have made these revisions:

First, in the comparison of GZnP1, GZnP2 and GZnP3 in **Fig 1c**, we changed to compare the sensor response at 60 seconds rather than at 180 seconds to show how each sensor would respond to transient increases in Zn²⁺. GZnP2 sensor response ($\Delta F/F_0$) increased from 0.42 to 0.83 in response to Zn²⁺ changing from 100 pM to 1 nM, while GZnP3 sensor increased response ($\Delta F/F_0$) from 0.21 to 0.94. Therefore, GZnP3 had a 2.25 fold greater response than GZnP2 to the same change in Zn²⁺ concentration.

Second, we compared GZnP3 with all published genetically encoded Zn²⁺ sensors and summarized the comparison in the new **Table 1**. We used the published dissociation constant (K_d) for each sensor to predict the sensor detection range. The K_d is defined as a concentration when the sensor is 50% saturated by Zn²⁺. When Zn²⁺ concentration changes within 10-fold below and above the K_d , these sensors can demonstrate the most significant changes (Park & Palmer, *Methods in molecular biology* 2014). As such, we defined the sensor detection range as the concentration from 10% to 10-fold of K_d and have listed all the detection range for each sensor in **Table 1**. For example, for our previously published GZnP2 sensor ($K_d= 352$ nM), the detection range it can mostly sensitively detect is 35 pM to 3.5 nM, which is consistent with our pDisplay assay that GZnP2 yields the highest sensitivity between 100 pM and 1 nM Zn²⁺ (**Fig. 1b & Fig. 1c**). Among 22 sensors listed in **Table 1**, 8 sensors with detection range overlapping with the range between 100 pM and 10 nM: ZapCY2, ZapCV2, e-CALWY-4, eCALWY-6, redCALWY-4, eZinCh-2, GZnP2 and GZnP3. From these sensors, GZnP3 demonstrated the highest dynamic range, while ZapCV2 and eZinCh-2 displayed the highest dynamic range for FRET-based Zn²⁺ sensors.

Third, we carried out further comparison between GZnP3 and ZapCV2, to compare and confirm the TRPML1-mediated Zn²⁺ signals in neurons. **Fig. 3h & Fig 3i** showed that ZapCV2 also detected TRPML1- mediated Zn²⁺ release, but GZnP3 displayed significantly higher sensitivity.

Comment 2: “Based on the prior evidence (from the co-author's lab) of zinc fluxes through the lysosomal ion channel TRPML1, the authors fuse GZnP3 to proteins resident in the lysosomes and test zinc fluxes in response to TRPML1 activation in several cell types. Similar to the prior evidence, TRPML1 activation by the pharmacological mean released zinc from the lysosome.” “the biological novelty of the studies presented here can be improved or explained better.”

Response: We appreciate the reviewer's concern and as suggested have provided a detailed explanation about biological significance of the paper in the introduction (pg. 1) and discussion (pg. 11). Our results have uncovered these novel biological findings: First, we developed tools

that can detect Zn^{2+} permeability of TRPML1 at relatively lower and physiological levels. Different from previous patch clamp method that has to express a plasma membrane targeting mutant TRPML1^{Va} in HEK293 cells and use 30 mM $ZnCl_2$ to elicit an inward current, we express wild type TRPML1 and used 99.7% lower Zn^{2+} concentrations (100 μ M $ZnCl_2$, which is in the physiological concentration range inside the vesicular lumen).

Second, prior studies only detected TRPML1-mediated Ca^{2+} release from lysosomes using the genetically encoded Ca^{2+} sensors. Here, we provide the first direct evidence that Zn^{2+} can be released from endolysosomes through TRPML1 channels in live neurons. It is well accepted that Zn^{2+} can be concentrated in the vesicles in secretory cells, however, it is not known whether and how Zn^{2+} can be released from vesicles to the cytosol. To our knowledge, our work provides the first detection of Zn^{2+} signals released from intracellular vesicles. Such discovery is very significant because it provides evidence that cellular Zn^{2+} signals are dynamic. The dynamic Zn^{2+} can play vital roles in regulating neuron functions (Please see the response to Reviewer 1 Comment 1 for detailed explanation).

Third, our work showed that TRPML1-mediated Zn^{2+} signals are unique in neurons and are especially higher in neurites. Given that TRPML1 mutations can cause the severe neurological disorder MLIV, and Zn^{2+} plays critical roles in mediating neuron function, our results suggest that lack of such Zn^{2+} signals might be involved in the pathological mechanisms of MLIV.

Fourth, we showed that TRPML1-mediated Ca^{2+} and Zn^{2+} signals are present in different cell types. For example, HeLa cells only possess TRPML1-mediated Ca^{2+} signals, but lack TRPML1-mediated Zn^{2+} signals. HeLa cells have been used in several studies to reveal the pathological roles of TRPML1 channels in MLIV. Using HeLa cells might overlook the involvement of Zn^{2+} in the function of TRPML1 and MLIV etiology.

Comment 3: “Some fusion constructs were more effective than others, but the source or meaning of these differences are not entirely clear.”

Response: We apologize for the lack of detailed clarification. We have included detailed discussion regarding the purpose and significance of these fusion constructs (pg. 7 - 8). One major advantage of genetically encoded sensors is that they can be targeted to different subcellular compartments by tethering with various targeting signal peptides or protein. However, fusion with a targeting motif can potentially alter the tertiary structure of the sensor to perturb the sensor response. We therefore analyzed the sensor response by fusion with different vesicular targeting proteins. We found that all fusion sensors except LAMP1-GZnP3 maintained the sensor sensitivity. Such results not only demonstrate the effects of different targeting motifs on sensor function, but also provide guidance on how to use and validate GZnP3 or other genetically encoded sensors when fusing with a targeting peptide.

Targeting sensors to microdomains can reveal information that are not detected by cytosolic sensors. For example, localized Ca^{2+} signaling microdomains are present close to the channels and membrane (Berridge, *Cell Calcium* 2006). The different constructs we made in this paper revealed several important findings. First, although high pools of Zn^{2+} are present in the synaptic vesicles (see review Frederickson et al., *J Nutr* 2000, **Supplementary Fig. 6c**), GZnP3 close to the presynaptic vesicles at axon terminals cannot detect significant Zn^{2+} signals when TRPML1 channels are activated (**Fig. 5d & 5e**), confirming that the TRPML1 cannot mediate Zn^{2+} release from the synaptic vesicles. Second, GZnP3-Rab7a displayed a rapid response to TRPML1-mediated Zn^{2+} release as compared to cytosolic GZnP3 and GZnP3-TRPML1 (**Fig. 5c & 5e**), suggesting that GZnP3-Rab7a can detect the localized Zn^{2+} signals in the microdomains near endolysosomes before equilibrium is reached. Third, GZnP3-TRPML1 and GZnP3-Rab7a can detect different pools of Zn^{2+} release. GZnP3-TRPML1 tends to detect Zn^{2+} release from lysosomes because it fails to detect Zn^{2+} signals when lysosomes were disrupted by GPN. Instead, GZnP3-Rab7a can still detect Zn^{2+} release from non-lysosomal vesicles (**Fig. 5f & 5g**).

Comment 4: The discussion of zinc role and physiology is somewhat bombastic. The authors claim that their work suggests that zinc sequestration or release could be important for endolysosomal maturation into degradative lysosomes; there is no evidence for this, or such evidence should be better discussed.

Response: We appreciate the reviewer's concern and have removed our language stating that Zn^{2+} release may be important for maturation. Rather we only keep the statement that suggests changes in lysosomal zinc may affect the degradative function of lysosomes (pg. 11).

Comment 5: The discussion of the buffering rate of calcium vs zinc is not supported by numbers.

Response: Calcium buffering mechanisms have been well studied. There are several cellular mechanisms that regulate reuptake Ca^{2+} into various compartments including the mitochondrial Ca^{2+} uniporter (MCU) and ER Ca^{2+} ATPase (SERCA), or exported from the cell through Plasma Membrane Ca^{2+} ATPase (PMCA) (see review Bagur & Hajnoczky, *Mol Cell* 2017). However, there has not been extensive research into the buffering rates of Zn^{2+} and therefore cytosolic Zn^{2+} dynamics are not well understood, especially in neurons.

In order to compare the buffering rate of Ca^{2+} vs Zn^{2+} , we loaded neurons with high Ca^{2+} and Zn^{2+} under similar conditions. When neurons were depolarized with high KCl, voltage-dependent Ca^{2+} channels (VGCCs) would open and allow both Ca^{2+} and Zn^{2+} entry (see review Inoue et al., *Curr Med Chem* 2015). As showed in **Supplementary Fig. 12**, Ca^{2+} can be quickly recovered to the baseline with 99% reduction from the peak levels at 15-minutes post depolarization, while Zn^{2+} levels were only reduced 12%.

To rule out the possibility that such difference was caused by sensor kinetic variations, we tested the *in situ* binding kinetics of GZnP3 as compared to GCaMP5 and demonstrated that GZnP3 does not have slower kinetics than GCaMP5 (**Supplementary Fig. 11**, please see Reviewer 1 Comment 4). Together, this data suggests that cytosolic Zn^{2+} cannot be buffered back to the baseline as quickly or effectively as Ca^{2+} .

Comment 6: The inverse relation between zinc and calcium handling can be explained by differences in transporter levels, which may be completely cell-type specific or even reflect the different course of adaptation to cell culture conditions.

Response: We agree with the reviewer's comments. Neurons have transporters that specifically load zinc into synaptic vesicles such as ZnT3 (Palmiter et al., *Proc Natl Acad Sci U S A* 1996) and SV31 (Barth et al., *J Neurochem* 2011). These transporters are not found in non-secretory cells. However, it is currently unknown exactly how Ca^{2+} and Zn^{2+} are concentrated into lysosomes in neurons. The inverse relationship of Ca^{2+} and Zn^{2+} is not just seen among different cell types, but also among various locations within the same neurons (soma versus neurites, **Fig. 6**). We do not know exactly why there are these differences in Zn^{2+} and Ca^{2+} handling, but think these are important and interesting findings that are overlooked by current studies that only focused on the involvement of Ca^{2+} signaling in the pathophysiology of MLIV. It is noteworthy that such a relation be reported between cell types, whereby further studies can determine its causes and/or function.

Due to space limitation, we could not include all these discussions and interpretations in the manuscript.

Reviewer 3:

Comment 1: "In the introduction part, the authors add the following references."

Response: We have included these references as suggested in the introduction (pg. 1).

Comment 2: “What was the effect of Na⁺、K⁺、Fe³⁺、Fe²⁺、Cr³⁺ etc. life elements ?”

Response: To address this question we repeated metal titration experiments *in vitro* and tested the sensor response to ZnCl₂, CaCl₂, MgCl₂, CuCl₂, MnCl₂, CoCl₂, NiSO₄, FeCl₂, KCl, FeCl₃, AlCl₃, and CrCl₃. Na⁺ was already included in the HEPES buffer (150 mM HEPES, 100 mM NaCl, 0.5 mM TCEP and 10% glycerol, pH 7.4) to maintain the ionic strength and make sure the sensor protein is at a soluble and native state. Zinc showed a statistically significant increase in fluorescence compared to all other metals evaluated.

Comment 3: “What is the effect of zinc citrate, zinc gluconate, etc?”

Response: We appreciate the reviewer’s comment. Zinc citrate (ZnCit) and zinc gluconate both are water soluble and therefore the labile Zn²⁺ cation dissociates from the citrate or gluconate anion. We believe that any water soluble compound containing Zn²⁺ will result in independent activity by the cation and anion components, whereby only the Zn²⁺ will interact with our sensor. For metal selectivity, we used ZnCl₂ to be consistent with other metal ions (such as CaCl₂, KCl, CuCl₂, FeCl₃, FeCl₂, CoCl₂,...). For all sensor sensitivity experiments (**Fig. 1b and 1c, Supplementary Fig. 2**) buffered zinc solutions were used as the source of labile Zn²⁺ ions (Please see Qin et al., *PNAS* 2011 for the detailed discussions how to make buffered Zn²⁺ solutions). Zinc gluconate was not available at chemical grade quality from our chemical suppliers, but we did test the effect of ZnCit on GZnP3. *In vitro* metal specificity experiments showed that there was a significantly greater response to ZnCit as compared to EDTA and no significant difference as compared to the ZnCl₂ response (see Figure below). From this data, we can assume that the Zn²⁺-associated anion has no appreciable impact on the response of GZnP3.

Reviewer 4:

Comment 1: “The relative usefulness of GZnP3 versus other indicators has not been clearly established.” “The overall dynamic range (no zinc to saturating zinc) is of low relevance for use of these indicators for physiological measurements given that all cells have a basal Zinc concentration and saturating zinc concentrations are unlikely to be encountered. More critical is the overall response amplitude within the range of signals investigated. Given this, it would be reasonable to expect a comparison of GZnP3 and a published indicator that has the largest dynamic range around the expected change of Zinc in neurons (100 pM – 1 nM according to the authors). These indicators could be compared in neurons in response to ML-SA1, showing both average traces (like **Fig. 3b** but just for the wild-type TRPML1) and quantification with bar graphs as done elsewhere.”

Response: We thank the reviewer for this suggestion. We agree that the dynamic range cannot predict the sensor's sensitivity for the physiological levels of Zn^{2+} dynamics. Instead, we should take into account both the sensor's apparent dissociation constant (K_d) for Zn^{2+} and dynamic range. The K_d is defined as a concentration when the sensor is 50% saturated by Zn^{2+} . When Zn^{2+} concentration changes between 10-fold of the K_d , the sensor can demonstrate the most sensitive response (Park & Palmer, *Methods in molecular biology* 2014). As such, we defined the sensor detection range as Zn^{2+} concentrations between 10-fold below and above K_d and have listed all the detection ranges for each sensor in **Table 1**. For example, our previously published GZnP2 sensor ($K_d = 352$ pM) can sensitively detect Zn^{2+} changes between 35 pM to 3.5 nM, which is consistent with the pDisplay results that GZnP2 yields the highest sensitivity within 100 pM -1 nM Zn^{2+} among the different concentration ranges tested (**Fig. 1b & Fig. 1c**). Among 22 sensors, we identified 7 previous sensors with detection range overlapping with the range between 100 pM and 1 nM: ZapCY2, ZapCV2, e-CALWY-4, eCALWY-6, redCALWY-4, eZinCh-2, GZnP2 and our new sensor GZnP3 (**Table 1**). From these sensors, GZnP3 demonstrated the highest dynamic range, while ZapCV2 and eZinCh-2 displayed the highest dynamic range for FRET-based Zn^{2+} sensors. As suggested, we made a further comparison between GZnP3 and ZapCV2, to compare and confirm the TRPML1-mediated Zn^{2+} signals in neurons. **Fig. 3h & 3i** showed that ZapCV2 also detected TRPML1-mediated Zn^{2+} release, but GZnP3 displayed significantly higher signals.

Comment 2: "Can calcium dynamics be complicating the interpretation of the results?"

Response: We apologize for the confusion in the metal selectivity results. For the metal selectivity assay, the control refers to the fluorescence of purified sensor protein and the EDTA group refers to the sensor protein treated with EDTA (12 μ M). EDTA treatment would reduce the baseline fluorescence because EDTA can chelate all the contaminated zinc during protein purification. For all metal treated groups, we first treated the sensor proteins with EDTA (12 μ M) to chelate all residual zinc, then added different cations (50 μ M) to test if the sensor can show any response to the extra levels of cations. The previous control group (used in the first submission) can be misleading. For better clarification of metal selectivity as compared to the apo state, we removed the control group and only presented the EDTA control in **Fig. 1d**.

In addition, our previously conducted *in situ* experiments tested the effect of elevated Ca^{2+} on the baseline fluorescence of GZnP3 in HeLa cells (see **Supplementary Fig. 1a**). This was performed using 5 μ M ionomycin treatment in the presence of 1 mM extracellular Ca^{2+} . Influx of Ca^{2+} showed no visible reduction of GZnP3 signal, suggesting that GZnP3 baseline signal is not perturbed by Ca^{2+} fluxes and is responding only to Zn^{2+} .

Comment 3: "Experiments in Figures 5 and 6 require further clarification"

"addition of ML-SA1 may not have been performed at steady state as GZnP3 signals were still decreasing when ML-SA1 was added. As a result, the response to ML-SA1 in the +GPN treatment could have been underestimated."

"In Figure 5b, the traces are also not at steady state when ML-SA1 is being added. This may be because some GZnP3-TRPML1 is on lysosomal membranes that are being lysed. These effects complicate the analysis of this experiment."

Response: We appreciate the reviewer's concerns. In order to address these concerns, we have repeated the imaging experiments in neurons treated with GPN. After 2 minute GPN treatment, we removed GPN from the dish and waited longer (at least 5 minutes) until the signals were returned to a steady state. We also presented the GPN pretreated imaging traces separately from ML-SA1 only treatments for better clarification. Of note, the ML-SA1 response following longer incubation after GPN was similar to our previous results with short-term washout presented in the first submission.

Comment 4: “Another concern is that since the GZnP3 domain in GZnP3-Rab7a is cytoplasmic, we would expect that its dF/F at time 300 sec (just before TPEN application) would be similar to that of GZnP3 (which is also cytosolic) in the +GPN condition (as well as in the -GPN condition).”

Response: We agree with the reviewer that after 300 sec treatment of ML-SA1, the Zn²⁺ signals detected by GZnP3 and GZnP3-Rab7a are expected to be similar. Intuitively, the Zn²⁺ concentrations within the endolysosomal microdomains and cytosol should reach equilibrium by buffering and diffusion after 300 sec. Unexpectedly, the Zn²⁺ signals detected by GZnP3-Rab7a at 300 sec were lower than the Zn²⁺ signals detected by GZnP3 (**Fig. 5e**). We have not fully understood the reasons for such differences. It is possible that Zn²⁺ signals in the microdomains near endolysosomal vesicles might be different from the cytosolic Zn²⁺ levels because of the presence of Zn²⁺ transport mechanisms that can uptake Zn²⁺ into endolysosomal vesicles. Previous studies have suggested that ZnT4 and TMEM163 might act as transporters in concert with TRPML1 to regulate Zn²⁺ flux between the cytoplasm and lysosomes (Kukic et al., *Biochem J* 2013; Cuajungco et al., *Traffic* 2014). Given that most of cellular Zn²⁺ is bound with protein and small ligands, the labile or accessible Zn²⁺ that can be detected by optical probes might also vary among different cellular compartments depending on uneven distributions of Zn²⁺ buffering ligands. It is also possible that the Rab7a can come off the membrane and diffuse out of punctate ROI we selected, which might explain why GZnP3-Rab7a displayed higher decay in response to ML-SA1 in cells with ruptured lysosomal membranes treated by GPN (**Fig.5f**). Also, we have to admit that one major limitation of our sensor is that it is intensity-based so that the signals can be influenced by sensor concentrations. The signals detected by GZnP3-Rab7a are punctate so that they are dimmer than the signals detected by the cytosolic GZnP3, which might also cause the variations. Further studies will be needed to clarify this phenomenon.

Comment 5: “The authors suggest there are no signals from synaptophysin-GZnP3 but there is a clear signal in Figure 5e. However, it’s not clear if it’s a real signal or an artifact due to application of the agonist.”

Response: We appreciate that the reviewer point this out. After reanalyzing the raw imaging data, we found that the small signal is due to the drifting of vesicles from 1 dish. We made a careful selection of new ROIs to avoid such artifacts. We compared the signals from synaptophysin-GZnP3 and other constructs 30 seconds after ML-SA1 addition, and the signals are significantly lower than the signals detected by other sensors (**Fig.5e**). Given that synaptic vesicles concentrate high levels of Zn²⁺ through ZnT3 transporter, the Zn²⁺ signals released from synaptic vesicles are expected to be higher (at least similar to the signals from endolysosomes). In addition, TRPML1 channel does not localize on synaptic vesicles based on our co-localization analysis (**Supplementary Fig. 6a & 6b, Fig.7c**). Therefore, such a small signal detected by synaptophysin-GZnP3 are negligible. Most importantly, this data demonstrates that high Zn²⁺ stored in synaptic vesicles are likely not released to the cytosol through TRPML1.

Comment 6: “Providing representative pictures of neurons in the various conditions (with GZnP3, GZnP3-TRPML1, and GZnP3-Rab7a) would be useful. In cases where GPN treatment was used, please provide pictures before treatment and at around time 0, before ML-SA1 was added.”

Response: We thank the reviewer for this suggestion and have included representative images in **Supplementary Figures 7, 8, and 9**.

Comment 7: “The authors analyze “fluorescent puncta”. Do they get similar results if they include the whole soma?”

Response: GZnP3-TRPML1 and GZnP3-Rab7a have generally punctate morphology, therefore ROIs were chosen to only include areas of the cell with sensor signals. The whole soma was not chosen in order to exclude areas of the soma where there was little to no sensor, which would negatively skew the average fluorescent intensity and not represent accurate sensor responses to each of the various treatments. For cytosolic GZnP3, ROIs comprised a large representative area of the soma.

Comment 8: “Finally, why is the response of GZnP3-Rab7a to ML-SA1 initially larger in the +GPN compared with the -GPN condition? See Figures 6c and i.”

Response: GPN is a substrate for the lysosome-specific enzyme Cathepsin C. Once cleaved by Cathepsin C, GPN becomes membrane impermeable and exerts an osmotic effect to disrupt the lysosomal membrane, resulting in the efflux of small peptides (Berg et al., *Biochem J* 1994) and ions such as Ca^{2+} and Zn^{2+} out of the lysosomes. These released ions might undergo reuptake into non-ruptured endosomes, thus generating a larger TRPML1-mediated ion release. Though we are unsure about the exact mechanisms by which this larger increase occurs, our main conclusion is that TRPML1-mediated Zn^{2+} release is consistently higher in neurites than in the soma.

Comment 9: Please discuss the potential caveats of using an TRPML1 overexpression system and its impact on the main results reported in this paper.

Response: We thank the reviewer for the suggestion. TRPML1 overexpression system has been greatly used to study the biological function of this channel because the ions released through the endogenous TRPML1 is too low (**Fig. 3g**). In order to detect the Ca^{2+} or Zn^{2+} signals microscopically, we overexpressed TRPML1 in neurons. But overexpression of a protein might perturb the cell metabolism and homeostasis. For example, studies have showed that TRPML1 overexpression can reduce neuronal death (Wang et al., *Oxidative medicine and cellular longevity* 2018). In addition, when TRPML1 protein expression levels are high, the membrane protein trafficking machinery that target TRPML1 to endolysosomal vesicles can be overwhelmed, thus causing mislocalization of TRPML1 in cells. Such mislocalization might cause Zn^{2+} release from non-endolysosomal vesicles. However, our co-localization experiments have showed that overexpressed TRPML1 does not localize on the synaptic vesicles, the major intracellular compartments storing high levels of Zn^{2+} in glutamatergic neurons (**Supplementary Fig. 6a & 6b**). In addition, using GZnP3 targeting on synaptic vesicles, we did not detect significant Zn^{2+} release from synaptic vesicles (**Fig. 5d & 5e**).

Comment 10: The 3rd paragraph of the discussion claims that this work demonstrates the measurement of *physiological* fluctuations in Zn^{2+} . Because TRPML1 was overexpressed and because an agonist was used, I don't think this claim can be made.

Response: In our introduction, we have discussed that the predicted physiological dynamics of Zn^{2+} refers to 100 pM - 1 nM changes. GZnP3 sensor can demonstrate significant higher sensitivity for Zn^{2+} in the range of 100 pM - 1 nM based on the pDisplay assay (**Fig. 1b & Fig. 1c**). In addition, we also provided evidence that GZnP3 can detect endogenous TRPML1-mediated Zn^{2+} release (**Fig. 3g**). The synthetic agonist ML-SA1 we used in this study gives similar or lower potency compared with the endogenous agonist $\text{PI}(3,5)\text{P}_2$ (Shen et al., *Nature Communications* 2012). So under the same concentration, the ML-SA1 induced signals are not larger than the signals induced by $\text{PI}(3,5)\text{P}_2$. For these reasons, we think we can still conclude that GZnP3 can detect physiological fluctuations in Zn^{2+} , but we add “between 100 pM and 1 nM” for clarification.

Comment 11: In Figure 1b, is the response of GZnP3 delayed compared with the application of the zinc buffers?

Response: Yes, there is a small delay because the pDisplay experiments are based on microscopy imaging with 10 s acquisition. It might have taken a while for the zinc to diffuse across the imaging dish. To confirm that GZnP3 does not have reduced kinetics, we also performed experiments *in situ* to test the turn on and turn off rates of GZnP3 as compared to GCaMP5. Please see **Supplementary Fig.11** and our response to Reviewer 1 Comment 4.

Comment 12: In the first subsection of the Results section, Supplementary Figure 1a should be Supplementary Figure 1b and vice-versa (or better, switch the figures in the Supplementary info).

Response: We thank the reviewer for catching this mistake. We have made the correction. In **Supplementary Fig. 1, 1a** and **1b** were swapped both in the illustration as well as the corresponding figure legend.

Comment 13: In Figure 2b, why is gray bar negative? Is this a pH effect?

Response: Yes, this is likely caused by pH. In HeLa cells, there is no Zn^{2+} that can be released, so the TRPML1-mediated H^+ release is likely being detected upon ML-SA1 addition without the presence of intracellular or extracellular Zn^{2+} . For further discussion, please see the supplementary methods, as well as **Supplementary Fig. 4 and 5** for discussion of pH effect.

Comment 14: Figure 3a, the response would be best represented as a change from the baseline i.e. we would suggest not to show the baseline image, but re-color the ML-SA1 and TPEN image to represent the % change from the baseline.

Response: We appreciate the reviewer's comment. However, we believe that it is more straightforward for readers to look at raw pseudocolored images at different treatments. We would like to avoid over-manipulation of the raw imaging data for the sake of reader comprehension. For readers not familiar with sensor work, it might be easier to understand the raw pseudocolored images (including the baseline image) to illustrate the changes in intensity seen in **Fig. 3b**. However, we have provided the suggested readjusted images (% changes to baseline) here below for the reviewer's reference (scale bar 20 μ M). Please note the numbers for the maximum and minimum are slightly different than as presented in **Fig. 3b** as the below image calculation does not take into account background subtraction.

Comment 15: In the Data Analysis section, it is written that “All t-tests took variance equality into consideration”. Please clarify. What was done if the samples did not show equal variance? Or did all samples have equal variance?

Response: We thank the reviewer for their concern, as the previous language was unclear. Samples were tested for variance using an F-test. T-tests were then performed with consideration for the determined variance equality or inequality. This has been updated in the methods section for statistics.

Reviewers' comments:

Reviewer #1 (Remarks to the Author):

The authors have adequately addressed all of this reviewer's concerns.

Reviewer #2 (Remarks to the Author):

The authors have addressed my concerns and I recommend this paper for publication.

Reviewer #3 (Remarks to the Author):

Authors done perfectly modification work, I am pleased to accept it for the publication.

Reviewer #4 (Remarks to the Author):

SUMMARY

We commend Minckley and colleagues for diligently attempting to address the concerns addressed by us and the other reviewers. We understand this took significant effort, especially given the number of reviewers and comments. With the new data, the article has addressed some concerns. We list below the remaining concerns.

MAJOR CONCERNS

A. The relative usefulness of GZnP3 versus other indicators has not been clearly established

We commend the authors for providing additional data to help validate the relative usefulness of GZnP3 versus other indicators. Here are remaining concerns:

(1) In the new table, the detection ranges quoted is misleading: the authors took the K_d and added an order of magnitude above or below. The ability to detect changes in Zinc will depend also on the steepness of the response curve, the response amplitude and kinetics, etc. The current range provides no additional information beyond the K_d and can be misleading. We therefore suggest removing this "detection range" (or to provide a better measure).

(2) We are unclear what is the current definition of "Dynamic range" in the table. We do not believe it is what we suggest here, as GZnP3 appears to have a ~5 fold response in that range based on Fig. 1c. Since the authors care about responses from 100 pM to 1 nM, then the table should list the response amplitude of each sensor for Zn concentration changes from 100 pM (thought to be the resting Zn^{2+} concentration in the cell) to 1 nM. Using published data from other sensors is acceptable. The authors should explain that is the measure used for FRET sensors? FRET sensors are typically characterized using the changes in the ratio of the acceptor/donor emission (dR/R).

(3) The authors are comparing GZnP3 against a ratiometric indicator which also uses FPs of different colors. This complicates the comparison. While figure 3h and 3i suggest a smaller response amplitude for ZapCV2, we are guessing the authors represented the response of ZapCh2 as the dF/F in a single channel (donor or emission) rather than the change in the ratio of the acceptor to the donor emission (dR/R). Moreover, since it uses non-circularly-permuted FPs, ZapCV2 may be much brighter and therefore overall may be able to detect Zinc transients as well as or better than GZnP3. The traces of ZapCV2 are much less noisy than those of GZnP3 – do the

authors know why this is? Given these considerations, it's not clear to us that GZnP3 is overall better than ZapCV2 (or eZinCh-2, which may be even better for the 100pm-1nm range). And overall Figure 3i suggests that ZapCV2 could possibly have been used for these experiments. This isn't necessarily a deal-breaker, but the authors should more clearly explain why their indicator is overall better.

B. Complications regarding the interpretation of the experiments utilizing GZnP3

We thank the authors for addressing many of our comments. Here are the remaining concerns regarding comments which were not fully addressed:

(4) Comment #4. We thank the authors for their reply. Is there any evidence that there exist microdomains around endolysosomal vesicles can impact local zinc concentration over such a long timescale (300 sec)? As a small ion, zinc is expected to diffuse on a much faster timescale. I believe that the existence of such semi-stable ionic microdomains within the cytosol has not been established and their demonstration would be a very significant discovery. If this is indeed the case, the authors should rephrase the corresponding paragraph to clarify that the existence of such zinc microdomains is highly speculative.

(5) Comment #5. When measuring at 30 sec, we agree with the authors that there is clearly a lower response of syn-GZnP3 to Zn²⁺ compared to other subcellular sensors. But in the first datapoint after adding ML-SA1, it is clearly higher than at baseline (Fig. 5d). Therefore, without showing that this is an artifact, I do not believe the author can claim that TRPML1 does *not* promote Zn²⁺ release from synaptic vesicles. The authors mentioned that TRPML1 channel does not localize on synaptic vesicles based on their co-localization studies. But Suppl. Fig. 6b shows a Pearson R of ~0.45-0.5, showing a correlation (albeit lower than the correlations with other subcellular markers). So while the release of Zn²⁺ from synaptic vesicles via TRPML1 may be proportionally less than in endosomes, the authors cannot claim it is undetectable. Making quantitative comparisons between subcellular areas is also tricky given that the behavior of at least GZnP3-Rab7a is not fully understood (see above).

(6) Comment #10. Figure 3g suggests that GZnP3 can detect zinc transients induced by a *strong* activator (ML-SA5). With ML-SA1, overexpression of TRPML1 is needed for detection. The conclusion is that GZnP3 *cannot* detect fluctuations of Zn²⁺ thought to occur naturally in neurons or other cell types i.e. it cannot detect "physiological" fluctuations. This statement doesn't invalidate the study and hopefully all conclusions remain valid when TRPML1 is not overexpressed. But it does suggest some caution in interpreting experimental results. The authors should therefore rephrase their claim that GZnP3 can detect physiological fluctuations of Zn²⁺.

MINOR POINTS

GZnP3 appears faster than GZnP2 (Figure 1b). This is an important characteristic and we would encourage the authors to consider highlighting this improvement.

Regarding comment 7, the average fluorescence intensity is irrelevant for single-FP sensors as one only measures the *change* in fluorescence. This is assuming of course that cellular autofluorescence has been corrected. However, we do not require the authors from reanalyzing their data using this method.

Response to Reviewers:

Reviewer 1: No issues to address

Reviewer 2: No issues to address

Reviewer 3: No issues to address

Reviewer 4: See below

Comment 1: In the new table, the detection ranges quoted is misleading: the authors took the K_d and added an order of magnitude above or below. The ability to detect changes in Zinc will depend also on the steepness of the response curve, the response amplitude and kinetics, etc. The current range provides no additional information beyond the K_d and can be misleading. We therefore suggest removing this “detection range” (or to provide a better measure).

Response: We agree with the reviewer that the sensor detection range is not solely dependent on the K_d and can also be affected by the steepness of sensor binding curve (hill coefficient), dynamic range (maximal signal to minimal signal) and kinetics. However a sensor demonstrates the largest sensitivity to the Zn^{2+} concentration ranging between 0.1 and 10 times of the K_d . We would like to change the definition of this range of Zn^{2+} concentration from “detection range” to “sensitive range”. When the range of our interest (100 pM - 1nM) is far away from the sensitive range, the sensor’s response is usually insignificant. For example, ZapCV5 sensor yields a very broad detection range as showed in the below binding curve, but the Zn^{2+} concentration range of interest (100 pM - 1nM) is far away from the sensitive range, and the sensor can barely show a response between 100 pM and 1nM Zn^{2+} .

We agree that this value can be misleading and have removed it from the table. But for most previously published genetically encoded sensors with very low dynamic range, their response to changes in Zn^{2+} out of their sensitive range is insignificant so that we would like to keep the discussion of this “sensitive range” in the introduction.

Comment 2: We are unclear what is the current definition of “Dynamic range” in the table. We do not believe it is what we suggest here, as GZnP3 appears to have a ~5 fold response in that range based on Fig. 1c. Since the authors care about responses from 100 pM to 1 nM, then the table should list the response amplitude of each sensor for Zn concentration changes from 100 pM (thought to be the resting Zn²⁺ concentration in the cell) to 1 nM. Using published data from

other sensors is acceptable. The authors should explain that is the measure used for FRET sensors? FRET sensors are typically characterized using the changes in the ratio of the acceptor/donor emission (dR/R).

Response: We appreciate the suggestion. The dynamic range here is defined as the ratio between the maximal signals to the minimum. Ideally, we would like to compare all sensor's response to 100 pM and 1nM Zn^{2+} in cells using the pDisplay assay that we used to compare GZnP1, GZnP2, and GZnP3. However, one disadvantage of FRET sensors is that their dynamic range (or amplitude) is significantly reduced when targeted onto the cell membrane due to restricted flexibility (Qin et al., *ACS Chem. Biol.* 2016). We have constructed pDisplay-ZapCV2 and pDisplay-ZapCV5 sensors, but their sensitivity is significantly reduced compared with the cytosolic version. We have also tried to collect and compare the *in vitro* sensor titration data from published work. Unfortunately, different readouts such as sensor occupancy (%), FRET ratio, and normalized ratio were used to report the sensor titration at different Zn^{2+} concentrations, which prevents a reliable comparison.

Instead, we would like to include a new parameter: the maximal Zn^{2+} response above baseline $[(S_{Zn} - S_0)/S_0]$, which is defined as the maximal changes in sensor signals in response to Zn^{2+} concentration larger than baseline concentration (~100 pM). For all the published sensors, they have been calibrated in cells in response to Zn^{2+} chelation (with Zn^{2+} chelator TPEN) and saturation (with Zn^{2+} /pyrithione), as demonstrated below for eZinCh2 (the below graph is taken from: Hessels et al., *ACS Chem. Biol.* 2015). The sensor signals (either FRET ratio, normalized ratio or fluorescent intensity) at baseline condition are denoted as S, while S_{Zn} indicates the signals when the sensor is saturated with Zn^{2+} . For eZinCh2 illustrated below, the baseline signal is 1 (R_0/R_0), and the Zn^{2+} saturation signal is 1.65 (R_{Zn}/R_0), so the maximal changes above the baseline is $(1.65-1)/1 = 0.65$.

With this parameter, we can compare the sensor sensitivity for changes in Zn^{2+} ranging from 100 pM to high Zn^{2+} . In addition, we can also rule out those sensors that have been saturated by baseline Zn^{2+} such as ZapCY1, eCALWY-2 and ZnGreen1 so that they yield no response to Zn^{2+} higher than baselines concentrations. For sensors with the similar K_d (~ 1 nM), this parameter could provide a reliable assessment about the differences in their sensor sensitivity from the baseline (100 pM) to nanomolar Zn^{2+} . We have calculated this parameter for all published sensors. For all the FRET sensors, their maximal response to Zn^{2+} saturation yield small signals (0-0.65) above or below the baseline, preventing them to sensitively detect changes in sub-nanomolar to nanomolar Zn^{2+} , while GZnP3 demonstrates significantly increased sensitivity to Zn^{2+} above the baseline concentration (**Table 1**).

As suggested by the reviewer, we have included a general note below **Table 1** to define S as signals, which is the readout used for different sensors. For FRET sensors, S refers to the FRET ratio (R), or normalized ratio (R/R_0); for intensity-based sensors, S refers to the fluorescence intensity (F).

Comment 3: The authors are comparing GZnP3 against a ratiometric indicator which also uses FPs of different colors. This complicates the comparison. While figure 3h and 3i suggest a smaller response amplitude for ZapCV2, we are guessing the authors represented the response of ZapCh2 as the dF/F in a single channel (donor or emission) rather than the change in the ratio of the acceptor to the donor emission (dR/R). Moreover, since it uses non-circularly-permuted FPs, ZapCV2 may be much brighter and therefore overall may be able to detect Zinc transients as well as or better than GZnP3. The traces of ZapCV2 are much less noisy than those of GZnP3 – do the authors know why this is? Given these considerations, it's not clear to us that GZnP3 is overall better than ZapCV2 (or eZinCh-2, which may be even better for the 100pm-1nm range). And overall Figure 3i suggests that ZapCV2 could possibly have been used for these experiments. This isn't necessarily a deal-breaker, but the authors should more clearly explain why their indicator is overall better.

Response: We are sorry that we did not label it correctly. When using the FRET sensor ZapCV2, the microscopic images were acquired in both FRET (Ex: 445 nm, Em: 535 nm) and CFP channel (Ex: 445 nm, Em: 480 nm), and the background corrected FRET and CFP intensities were used to calculate the FRET ratio (R). We converted R to $\Delta R/R_0$ to be consistent with GZnP3. We have corrected the label and included a detailed description about the FRET imaging in methods.

The traces of ZapCV2 are less noisy because it is a ratiometric sensor that any noise signals (from light, camera, or tiny cell drift) can be normalized. ZapCV2 is not chosen for this study due to two reasons. First, ZapCV2 shows only a 6% increase in signal, which is significantly lower than GZnP3 (**Fig. 3i**). Second, when ZapCV2 is attached to the cell membrane, its sensor response is significantly diminished, while the GZnP sensor can still maintain the similar sensitivity (as shown in the below figure from: Qin et al., ACS Chem. Biol. 2016). This is a common weakness for FRET-based sensors that attachment with a targeting motif can significantly compromise their sensitivity. For example, when eZinCh-2 is targeted to the ER, its dynamic range is reduced from approximately 2.6 to 1.3 (Hessels et al., ACS Chem. Biol. 2015). For this reason, targeting ZapCV2 to the endolysosomal membrane would limit its ability to detect Zn^{2+} dynamics in the endolysosomal microdomains. Instead, we provided evidence in this work that targeting GZnP3 to TRPML1 channel, endolysosomal vesicles, and synaptic vesicles did not affect the GZnP3 sensor sensitivity so that it can be used to monitor the localized Zn^{2+} signals close to endolysosomal vesicles (**Fig. 4**).

Comment 4: Is there any evidence that there exist microdomains around endolysosomal vesicles can impact local zinc concentration over such a long timescale (300 sec)? As a small ion, zinc is expected to diffuse on a much faster timescale. I believe that the existence of such semi-stable ionic microdomains within the cytosol has not been established and their demonstration would be a very significant discovery. If this is indeed the case, the authors should rephrase the corresponding paragraph to clarify that the existence of such zinc microdomains is highly speculative.

Response: Our data did not support that the Zn^{2+} concentration in the local microdomain can last over a long timescale (300 s). Instead, we showed that the local signals as detected by GZnP3-Rab7a were significantly higher than global cytosolic signals as detected by GZnP3 initially (within 30s). At 300s, the signals detected by GZnP3-Rab7a were reduced and cytosolic signals were increased because Zn^{2+} ions diffuse from the site at which they were released to the whole cytosol. Our work supports that Zn^{2+} microdomains might exist around the endolysosomal vesicles that possess a localized high Zn^{2+} concentration transiently (<20 s) (**Fig. 5c**) upon TRPML1 channel activation. Given that there is a delay between Zn^{2+} release and agonist addition (the agonist must diffuse throughout the dish, cross the cell plasma membrane, and then act on the channel), the induced high Zn^{2+} signals in the endolysosomal microdomain should be induced less than 20 seconds after channel opening. At 300 seconds, the signal has been stabilized because the local Zn^{2+} concentration has equilibrated to cytosolic levels. Unlike Ca^{2+} which can be buffered quickly back to the baseline (**Fig.6 & Supplemental Fig. 12**), Zn^{2+} has slower buffering mechanisms that may be responsible for the delayed Zn^{2+} decay and lack of return to baseline concentrations. We have included a statement (highlighted in yellow, pg 8 & 11) in the paper to discuss this speculation.

Comment 5: When measuring at 30 sec, we agree with the authors that there is clearly a lower response of syn-GZnP3 to Zn^{2+} compared to other subcellular sensors. But in the first datapoint after adding ML-SA1, it is clearly higher than at baseline (Fig. 5d). Therefore, without showing that this is an artifact, I do not believe the author can claim that TRPML1 does *not* promote Zn^{2+} release from synaptic vesicles. The authors mentioned that TRPML1 channel does not localize on synaptic vesicles based on their co-localization studies. But Suppl. Fig. 6b shows a Pearson R of ~0.45-0.5, showing a correlation (albeit lower than the correlations with other subcellular markers). So while the release of Zn^{2+} from synaptic vesicles via TRPML1 may be proportionally less than in endosomes, the authors cannot claim it is undetectable. Making quantitative comparisons between subcellular areas is also tricky given that the behavior of at least GZnP3-Rab7a is not fully understood (see above).

Response: As requested by the reviewer, we have changed our language to state that the majority of TRPML1-mediated Zn^{2+} appears to come from endolysosomal compartments (pg 11). However, we still think that we lack enough evidence to support that TRPML1 can promote Zn^{2+} release from synaptic vesicles.

Regarding the co-localization analysis, interpretation of Pearson's value around 0.4 is challenging and controversial (Dunn, Kamocka, and McDonald, *Am J Physiol Cell Physiol*. 2011). Statistically, Pearson's value between 0.3 and 0.5 is considered relatively weak (David S. Moore, William I. Notz, et al. *The Basic Practice of Statistics*. 2012).

At a typical optical resolution of a confocal microscope (XY: 200–300 nm, Z: 500–575 nm), studies have found that the extent of colocalization is typically overestimated, especially at high molecular densities (Xu et al., *FEBS Journal* 2016). The false positive correlation is caused by random co-localization between intracellular dynamic molecules and low resolution of current

light microscopes that cannot distinguish the spatial autocorrelation (the value of one pixel is likely to be similar to that of its neighboring pixels) from real biological association.

To provide a more reliable quantification, a negative control should be performed to compare pairs of images with only random co-localization using the same set of microscopy imaging and imaging analysis system (Dunn, Kamocka, and McDonald, *Am J Physiol Cell Physiol.* 2011). We have run a negative control looking at the colocalization of mCherry-TRPML1 and mito-EGFP (mitochondrial marker). There is clearly no co-localization between TRPML1 and mitochondria because TRPML1 protein has no mitochondrial localization signals and the TRPML1 marked structures are punctate, showing drastically different morphology than mitochondria. However, they still show autocorrelation and random colocalization with an average Pearson's R Value of 0.42 ± 0.059 ($n = 6$). Please see the representative images below and **Supplementary Fig. 6a & b**. Compared with the negative control, the correlation between TRPML1 and synaptic vesicle markers is not statistically significant so that we can conclude that TRPML1 channel is predominately localized on endolysosomal vesicles, not synaptic vesicles (pg. 6).

Given the tiny signals detected by GZnP3 sensor targeted on synaptic vesicles that are not proportional to the high concentrations of labile Zn^{2+} present inside the synaptic vesicles and the poor co-localization of TRPML1 with synaptic vesicle marker, we would like to conclude that “the majority of TRPML1-mediated Zn^{2+} signals are not released from synaptic vesicles” (pg 8).

Comment 6: Figure 3g suggests that GZnP3 can detect zinc transients induced by a *strong* activator (ML-SA5). With ML-SA1, overexpression of TRPML1 is needed for detection. The conclusion is that GZnP3 *cannot* detect fluctuations of Zn^{2+} thought to occur naturally in neurons or other cell types i.e. it cannot detect “physiological” fluctuations. This statement doesn't invalidate the study and hopefully all conclusions remain valid when TRPML1 is not overexpressed. But it does suggest some caution in interpreting experimental results. The authors should therefore rephrase their claim that GZnP3 can detect physiological fluctuations of Zn^{2+} .

Response: We agree to change the “physiological” to “subnanomolar” (pg 11).

Comment 7: GZnP3 appears faster than GZnP2 (Figure 1b). This is an important characteristic and we would encourage the authors to consider highlighting this improvement.

Response: We appreciate this comment and have included a new supplementary figure (**Supplementary Figure 2b & 2c**) to highlight the faster kinetics of GZnP3 as compared to GZnP2. We included a new sentence in results to highlight the improved kinetics (pg 3).

REVIEWERS' COMMENTS:

Reviewer #4 (Remarks to the Author):

The authors have sufficiently addressed all concerns. We thank the authors for being responsive to our concerns and suggestions.